# Glycerol as Ligand in Metal Complexes—A Structural Review

Laurent Plasseraud 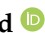

Institut de Chimie Moléculaire de l'Université de Bourgogne (ICMUB), UMR-CNRS 6302,
Université de Bourgogne, 9 Avenue A. Savary, F-21078 Dijon, France; laurent.plasseraud@u-bourgogne.fr

**Abstract:** The molecule glycerol ($H_3$gly) plays a key role in sustainable and green chemistry. Having been discovered for over 200 years and produced from vegetable oils and animal fats by hydrolysis, saponification and transesterification reactions, this natural triol is today employed in a wide range of cosmetic, food, polymer and pharmaceutical applications. Moreover, it is an essential C3 precursor in the chemical industry, used in the production of several intermediates and it avoids the need for petroleum-based precursors. Less famous but just as exciting, in the domain of coordination chemistry, glycerol is also proving to be a suitable ligand, capable of binding to one or more metal centres, either directly in its triol $H_3$gly form (rather rare), or in its various deprotonated glycerolate forms, such as $[H_2gly]^-$, $[Hgly]^{2-}$ and $[gly]^{3-}$ (in most cases). Since the 1970s, various molecular structures prepared from glycerol and metallic and organometallic precursors, ranging from mononuclear complexes to sophisticated aggregates and coordination polymers, have been isolated and characterised. On the basis of the single-crystal X-ray diffraction structures described so far in the literature and deposited in the Cambridge Structural Database, in this structural inventory, we review the different modes of coordination of glycerol and glycerolates with metals.

**Keywords:** bio-based ligand; coordination chemistry; metal glycerolate; single-crystal X-ray analysis; crystallographic structures

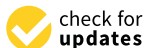



## 1. Introduction

Glycerol ($H_3$gly), also known as glycerine (IUPAC name: propane-1,2,3-triol), is a natural C3 molecule with three alcohol functions, two primary and one secondary. In nature, this triol is abundant in the form of triglycerides (vegetable oil) and is also present in the phospholipid skeleton (cell wall) (Scheme 1). The role of glycerol and its derivatives in the biochemistry of living organisms has been the subject of a recent review [1]. Historically, glycerol was accidentally discovered in 1783 by Carl Wilhelm Scheele, a Swedish–German chemist, by boiling olive oil with lead oxide. In 1823, in France, Michel-Eugène Chevreul clearly established its formation during the saponification of triglycerides and reported this discovery in *Recherches chimiques sur les corps gras d'origine animale* [2]. The empirical formula for glycerol, $C_3H_8O_3$, was finally determined by Pelouze in 1836. Almost fifty years later, Berthelot and Lucea definitively identified its structural formula as $C_3H_5(OH)_3$ [3]. Physically, glycerol is a viscous colourless liquid, odourless, sweet-tasting, highly hygroscopic and non-toxic. It can dissolve in polar solvents and is miscible in water and ethanol; however, it is insoluble in aromatic and halogenated solvents. Chemically, glycerol can be isolated as the main co-product of hydrolysis, saponification and transesterification of triglycerides from vegetable oils, animal fats and waste oils, leading to the formation of fatty acids, soaps and esters, respectively. The most popular application of glycerol is probably its use in the composition of nitroglycerine and its stabilisation in the form of dynamite sticks, invented by Nobel, but glycerol is widely used industrially in various sectors, especially for food, cosmetics, pharmaceuticals and polymer applications. In 2018, the worldwide production of crude glycerol was estimated at 3.8 million tonnes, mainly from the production of bioesters from vegetable oils [4]. Glycerol can also be used advantageously as a reaction solvent for organic synthesis and catalytic applications. It is then

considered a green solvent [5]. It is known, for example, to significantly improve the yield of certain aza-Michael reactions [6]. Glycerol can also be employed as a building block in the synthesis of bio-based dendrimers [7] and ionic liquids [8]. In the field of renewable resources, glycerol is considered to be an important sustainable platform molecule produced from the oil biorefinery, which can then be converted into useful chemical products, offering alternatives to conventional petroleum-based routes [9]. Epichlorohydrin, acrolein, 1,3-propanediol, 1,2-propanediol, glycerol carbonate and glycidol are a few examples of essential chemical synthons that can be produced industrially from glycerol [10].

|       |       |       |
|-------|-------|-------|
| (a)   | (b)   | (c)   |

**Scheme 1.** Molecular representations of: (**a**) glycerol (H$_3$gly), (**b**) triglyceride and (**c**) phospholipide (the C3 skeleton is shown in bold).

Metal glycerolates (M-glycerolate), recently defined by Gonçalves and Shahbazian-Yassar as *metal alkoxides consisting of stacked layers of glycerol coordinated to metal ions* [11], have aroused a growing interest because they can be used as catalytic systems, especially for the production of biodiesel and glycerol carbonate through transesterification processes [12–14]. They are also suitable precursors for producing nanoparticles and nanostructure materials for technical and biomedical applications [15]. Recently, Khonina's group demonstrated the antimicrobial activity of biogenic element glycerolates [16,17]. From a synthetic point of view, metal glycerolates are generally prepared by reacting metal precursors of various types, such as oxides, hydroxides and halides, with glycerol, at high temperatures of up to 245 °C. To date, several metal glycerolates have been isolated as single crystals and their structures were determined by X-ray crystallographic diffraction, revealing numerous coordination modes to metal centres. Glycerol and its deprotonated forms act as monodentate, bidentate, tridentate and bridging ligands. (Scheme 2). Based on hits collected on the Cambridge Structural Database (CSD), we report herein an inventory of these structures, focusing successively on these different types of coordination. To our knowledge, however, there are only a few examples of coordination metal complexes directly exhibiting the H$_3$gly glycerol adduct. Most of the X-ray structures identified contain deprotonated forms of the glycerol molecule, which correspond to [H$_2$gly]$^-$, [Hgly]$^{2-}$ and [gly]$^{3-}$ glycerolato ligands. While the glycerol adduct is only coordinated to the metal by one or more alcoholic M−O bonds, deprotonated ligands can also involve alkoxide-type bonds to connect to one or more metal centres, which in many examples favours the formation of polynuclear species.

**Scheme 2.** Possible coordination modes of glycerol and glycerolato ligands to one and several metal centres (M).

## 2. Materials and Methods

Regarding the general method used to carry out this structural inventory, the successive crystals described below were identified from quests on the online portal of the Cambridge Structural Database (CSD) web interface (version 2023) [18]. The last request to prepare this survey was carried out in December 2023. The selected and appropriate cif files were exported and reworked in MERCURY CSD (version 2020.3.0 [19]) in order to highlight the coordination modes of the glycerol and glycerolato ligands to metal centres. Arbitrarily, we chose to consider the crystals with $H_3gly$ first, then $[H_2gly]^-$, $[Hgly]^{2-}$ and finally $[gly]^{3-}$ ligands. However, some compounds display several distinct coordination modes within their structure. In this case, each coordination mode will be described in its respective section. In addition to the structural description, comparison tables including selections of structure are also displayed at the end of each section (Tables 2–5). Experimental details of how to synthesise and obtain crystals of the described compounds are also included and, where available in the original publications, some relevant analytical data (infrared and NMR spectroscopy, mass spectrometry, degradation temperature) resulting from the coordination of glycerol and/or glycerolate are also specified.

## 3. Results and Discussion

### 3.1. Direct Coordination of Glycerol to Metal Centres, as $H_3gly$ Adduct

The direct coordination of the glycerol molecule to a metal centre, such as an $H_3gly$ ligand, has rarely been described up to now in the literature. To our knowledge, only two previous studies have reported the characterisation of such species in a solid state [20,21]. In 2003, Prior and Rosseinsky studied the control of interpenetration and chirality of a family of metal–organic frameworks (MOFs), with the aim of examining, in particular, the role of auxiliary ligands (alcohols and aromatic amines) [20]. In this context, they isolated new MOFs of formula $Ni_3$(1,3,5-benzenetricarboxylate)$_2$(py)$_6$(glycerol) (**1**) in which the molecule of glycerol can be observed as a $H_3gly$ ligand (py = pyridine). Crystals of **1** were obtained by vapour diffusion at room temperature using a multilayer protocol. Interestingly, the authors showed that in **1,** the glycerol molecule occupies two distinct coordination modes with respect to the nickel atom, being both bidentate and monodentate in proportions of 45:55, respectively (Figure 1). According to the authors, monodentate coordination is favoured because it promotes the formation of hydrogen bonds. In both conformations, the geometry of the nickel atom is kept octahedral. The study also includes the design and characterisation by X-ray crystallographic analysis of additional interpenetrating networks using another aromatic amine (pyridine, 4-picoline) and several alcohols (ethylene glycol, 1,2-propanediol, 1,4-butanediol, meso-2,3-butandiol, and 1,2,6-hexanetriol).

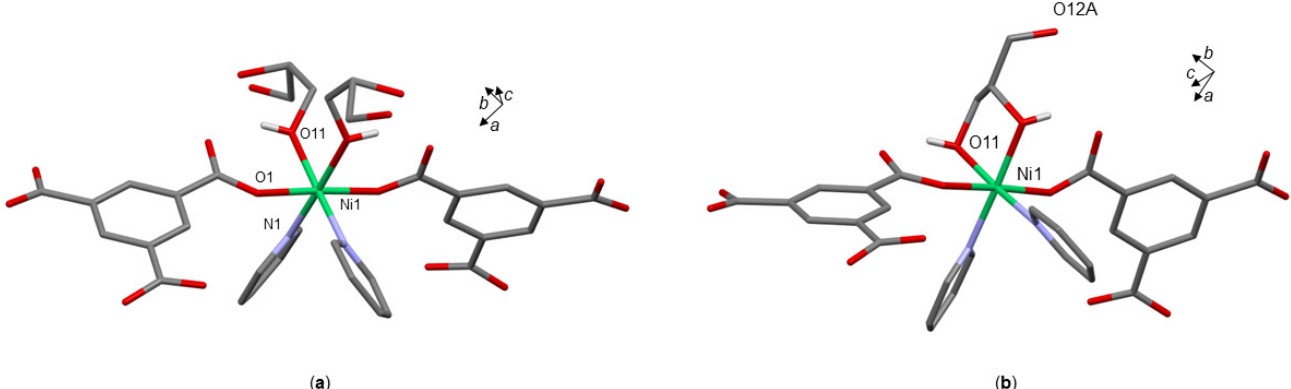

**Figure 1.** View of the coordination of the nickel atom in **1** showing the possible binding of the glycerol molecule through: (**a**) monodentate and (**b**) bidentate links (MERCURY representation, adapted from [20]). Only the hydrogen atom of the OH group bonded to the nickel atom could be located (colour code: green—nickel, red—oxygen, blue—nitrogen, grey—carbon, white—hydrogen).

Other examples of $H_3$gly ligands were reported a few years later, in 2006, but they were taken from a single study. Indeed, Naumov, Kim and coworkers described the synthesis and characterisation of six lanthanide salts, isolated as single crystals and consisting of $[Re_6Q_8(CN)_6]^{4-}$ anions combined with binuclear $[Ln_2(H_2gl)_2(H_3gly)_4]^{4+}$ cations (**2**: Ln = La, Q = S; **3**: Ln = Nd, Q = S; **4**: Ln = Gd, Q = S; **5**: Ln = La, Q = Se; **6**: Ln = Nd, Q = Se; **7**: Ln = Gd, Q = Se) [21]. Crystals of compounds **2**–**7** were obtained using the same experimental procedure, first by treating an aqueous solution of lanthanides chlorides with KOH and glycerol, and then by adding a solution of $K_4[Re_6S_8(CN)_6]$ in $H_2O$. The mixture was then boiled until the precipitation and crystallisation of compounds **2**–**7**. Yields ranged from 55 to 70%. All the salts crystallised in the triclinic $P\bar{1}$ space group and were iso-structural. All $[Ln_2(H_2gl)_2(H_3gly)_4]^{4+}$ cations exhibit a centrosymmetric binuclear structure. Remarkably, the lanthanide atoms of each cation bear two glycerol molecules as chelating ligands but display two distinct coordination modes. One is bidentally linked, while the second is tridentate (Figure 2, right). The two lanthanide atoms are also linked by two bridging $[H_2gly]^-$ ligands, the coordination of which will be developed more in detail in the next section and shown in Figure 4. Ln(III) atoms are non-coordinated and exhibit a distorted tricapped trigonal prism geometry environment. The non-coordinated OH groups of the glycerol and glycerolate ligands are involved in intermolecular hydrogen bonding with the nitrogen atoms of the CN groups of the $[Re_6Q_8(CN)_6]^{4-}$ anions, leading to the expansion of a three-dimensional network (Figure 3). Compounds **2**–**7** were also characterised by mass spectrometry and infrared spectroscopy and their magnetic properties were evaluated. La-based crystals (**2** and **5**) are diamagnetic, while Nd- and Gd-based (**3**, **4**, **6** and **7**) are paramagnetic. For compound **2**, electrospray ionisation positive mode corroborated the X-ray structure by revealing the presence of an intense signal at $m/z$ = 207.2 corresponding to the $[La_2(\mu-H_2gly)_2(H_3gly)_4]^{4+}$ fragment. In addition, infrared spectroscopy appears to be well suited to establishing the coordination of glycerol onto metal centres. This is because the $\nu_{C-O}$ and $\delta_{C-OH}$ characteristic bands are, respectively, shifted to lower frequencies and split, compared to pure glycerol (Table 1).

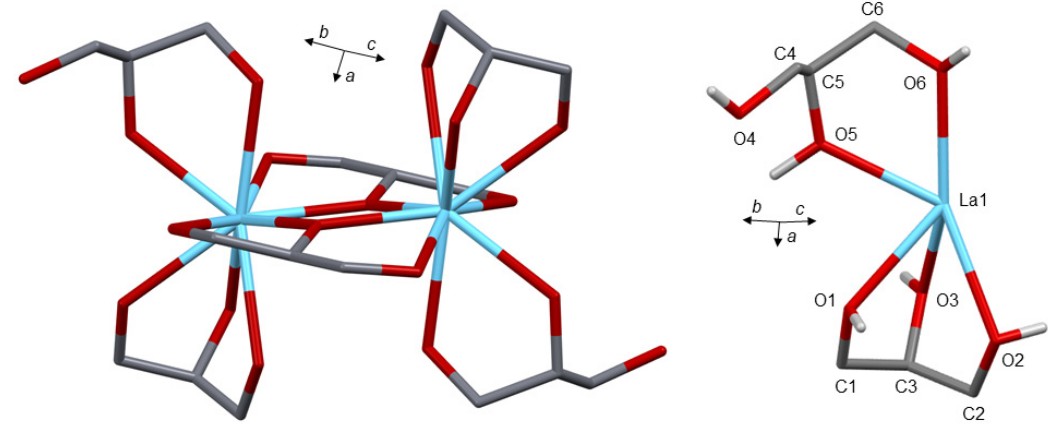

**Figure 2. Left**: general view of the skeleton of **2** (MERCURY representation, adapted from [21]). $[Re_6Q_8(CN)_6]^{4-}$ anion and hydrogen atoms are omitted for clarity (colour code: sky blue—lanthane, red—oxygen, grey—carbon, white—hydrogen). **Right**: detail of the coordination of the La atom by the two $H_3$gly ligands highlighting bidentate and tridentate bindings. Only the hydrogen atoms of OH groups are shown.

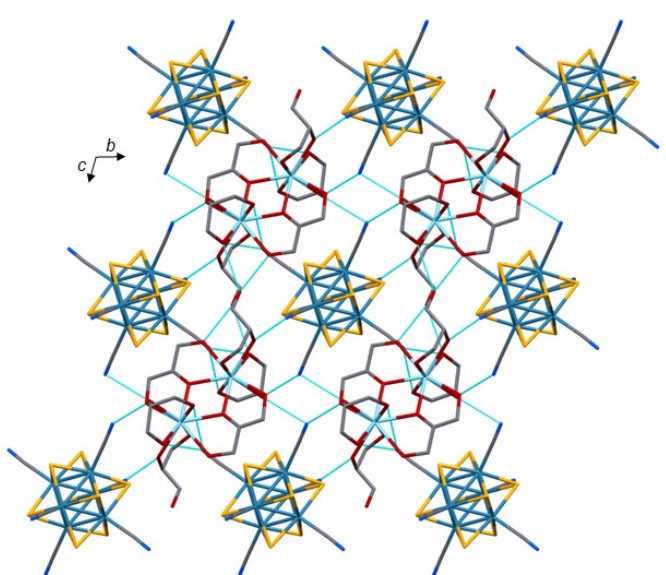

**Figure 3.** Crystal packing of **2** (MERCURY view, adapted from [21]). Hydrogen atoms are omitted for clarity. Hydrogen bonds are shown by light blue dotted lines.

**Table 1.** Selection of characteristic infrared bands (FTIR) showing the influence of coordination.

| Compound/Crystal | Coordination Mode of Glycerol and Glycerolate | $\nu_{O-H}$ (cm$^{-1}$) | $\nu_{C-O}$ (cm$^{-1}$) | $\delta_{C-OH}$ (cm$^{-1}$) | Ref. |
|---|---|---|---|---|---|
| glycerol [a] | / | 3345 | 1109<br>1034 | 1331 | [21] |
| **2** [a] | $H_3$gly, $[H_2gly]^-$ | 3500–3250 | 1095<br>1029 | 1361<br>1348<br>1334 | [21] |
| **3** [a] | $H_3$gly, $[H_2gly]^-$ | 3500–3250 | 1095<br>1028 | 1359<br>1346<br>1332 | [21] |
| **4** [a] | $H_3$gly, $[H_2gly]^-$ | 3500–3250 | 1095<br>1031 | 1359<br>1346<br>1332 | [21] |

**Table 1.** *Cont.*

| | | | | | |
|---|---|---|---|---|---|
| **5** [a] | $H_3gly$, $[H_2gly]^-$ | 3500–3250 | 1095<br>1031 | 1359<br>1346<br>1332 | [21] |
| **6** [a] | $H_3gly$, $[H_2gly]^-$ | 3500–3250 | 1095<br>1029 | 1359<br>1346<br>1332 | [21] |
| **7** [a] | $H_3gly$, $[H_2gly]^-$ | 3500–3250 | 1095<br>1028 | 1357 | [21] |
| **8** [a] | $[H_2gly]^-$ | 3400 | NS | NS | [22] |
| **9** [a] | $[H_2gly]^-$ | 3200 | NS | NS | [23] |
| **14** [a] | $[Hgly]^{2-}$ | 3410 | 1110<br>1063 | NS | [24] |
| **20** [b] | $[Hgly]^{2-}$ | 3208 | NS | NS | [25] |
| **21** [c] | $[Hgly]^{2-}$, $[gly]^{3-}$ | 3641 | 1138<br>1080 | 1380<br>1315 | [26] |
| **22** [a] | $[Hgly]^{2-}$ | 3399 | 1171<br>1027 | 1384 | [27] |
| **30** [b] | $[gly]^{3-}$ | absent | 1100–1000 | absent | [28] |

Samples analysed in [a] transmission, [b] attenuated total reflection (ATR), [c] diffuse reflectance (DR) mode; NS: not specified.

**Table 2.** Comparison of selected structural parameters relevant to the coordination of $H_3gly$ in crystals **1**–**7**.

| Crystal | M−O(alcoholic) (Å)<br>Bidentate Mode | M−O(alcoholic) (Å)<br>Tridentate Mode | M−O(alcoholic)−C (Deg)<br>Bidentate Mode | M−O(alcoholic)−C (Deg)<br>Tridentate Mode | CSD Entry<br>Deposition<br>Number | Ref. |
|---|---|---|---|---|---|---|
| **1**<br>M = Ni | 2.088 | | 105.5 | | HUYKUH<br>207688 | [20] |
| **2**<br>M = La | 2.541(7)<br>2.5730 | 2.554(7)<br>2.616(9)<br>2.690(8) | 116.8<br>125.3(5) | 114.0(7)<br>118.1(7)<br>121.1(7) | VEBYIL<br>269462 | [21] |
| **3**<br>M = Nd | 2.484(6)<br>2.559(5) | 2.487(6)<br>2.541(7)<br>2.637(6) | 117.6(4)<br>124.8(4) | 113.9(5)<br>116.9(5)<br>120.9(5) | VEBYOR<br>269463 | [21] |
| **4**<br>M = Gd | 2.439(5)<br>2.5632 | 2.457(5)<br>2.519(5)<br>2.601(5) | 116.9<br>125.2(4) | 113.5(4)<br>117.8(4)<br>120.0(4) | VEBYUX<br>269464 | [21] |
| **5**<br>M = La | 2.546(9)<br>2.6354 | 2.60(1)<br>2.6354<br>2.71(1) | 115.1<br>124.2(6) | 115.1<br>118.9(9)<br>119(1) | VEBZAE<br>269465 | [21] |
| **6**<br>M = Nd | 2.474(6)<br>2.6275 | 2.504(7)<br>2.547(8)<br>2.663(7) | 114.9<br>125.4(5) | 112.7(6)<br>119.0(7)<br>118.9(6) | VEBZEI<br>269466 | [21] |
| **7**<br>M = Gd | 2.42(1)<br>2.6312 | 2.44(1)<br>2.50(1)<br>2.60(1) | 115.1<br>125.5(7) | 112(1)<br>118(1) | VEBZIM<br>269467 | [21] |

### 3.2. $[H_2gly]^-$ Coordination Mode of Glycerolato Ligand to Metal Centres

As indicated previously in Section 3.1, in addition to $H_3gly$ adducts, compounds **2**–**7** also contain two $[H_2gly]^-$ ligands. They chelate and bridge the two lanthanide atoms. The result is the formation of a planar $Ln_2O_2$ central four-membered ring involving the $O^-$ moiety of $[H_2gly]^-$ acting as bridging ligands (Figure 4).

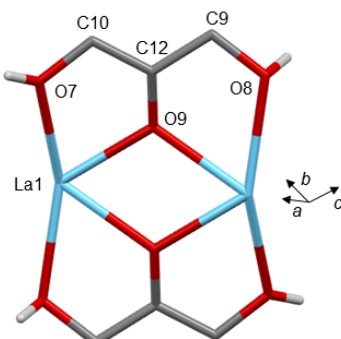

**Figure 4.** View of the coordination of the La atoms of **2** by two bridging [H$_2$gly]$^-$ ligands (MERCURY representation, adapted from [21]). Only the hydrogen atoms of OH groups are shown. H$_3$gly ligands were omitted as they are already shown in Figure 2 (colour code: sky blue−lanthane, red−oxygen, grey−carbon, white−hydrogen).

In 1998, Chakravorty and coworkers reported the synthesis and structure of vanadate esters of glycerol and propane-1,3-diol [22]. They established, in particular, the first X-ray structure of an oxovanadium alkoxide bearing a glycerolato ligand by characterising the mononuclear complex [VO(L)(H$_2$gly)] (**8**) (L = hydroxyphenylmethylenehydrazone of salicylaldehyde) (Figure 6). From a synthetic point of view, complex **8** was prepared by adding an excess of glycerol to a methanolic solution of VO(acac)$_2$ (acac = acetylacetonate) and H$_2$L, mixed under stoichiometric conditions and then warmed in ambient air for ten minutes. Single crystals suitable for an X-ray crystallographic analysis were collected from the resulting dark-coloured solution. Compound **8** crystallises in the monoclinic *P*21/*n* space group.

The glycerolato ligand is bidentately coordinated to the vanadium atom forming a five-membered V(*O*,*O*) chelate ring. The V−O distances exhibit distinct values, corresponding to 1.794 (6) and 2.308 (8) Å, which are attributed to the V−O(alkoxide) and V−O(alcoholic) bonds, respectively. The L$^{2-}$ ligand displays a tridentate ONO coordination leading to five- and six-membered rings. The vanadium atom occupies a distorted octahedral geometry. In the crystal lattice, **8** is organised into infinite chains extending along the *c*-axis, via intermolecular hydrogen bonds involving the non-coordinated OH and NH groups of [H$_2$gly]$^{2-}$ and L$^{2-}$, respectively (Figure 5). Two other compounds similar to **8** were also synthesised using H$_2$L, the hydroxyphenylmethylenehydrazone of 4-hydroxy-4-phenylbut-3-en-2-one and 2-hydroxynaphthaldehyde, respectively. It is interesting to note that when the reactions occur in the presence of propane-1,3-diol (H$_2$pd), instead of glycerol, the coordination of Hpd to the vanadium atom leads to the formation of a six-membered ring with a distorted chair conformation. From a spectroscopic point of view, the authors have shown that the size of the five- and six-membered V(*O*,*O*) chelate ring can be discerned from the chemical shifts recorded by $^{51}$V NMR, which show a difference of around 30 ppm ([VO(L)(H$_2$gly)] (**8**): −509 ppm, ([VO(L)(Hpd)]: −538 ppm).

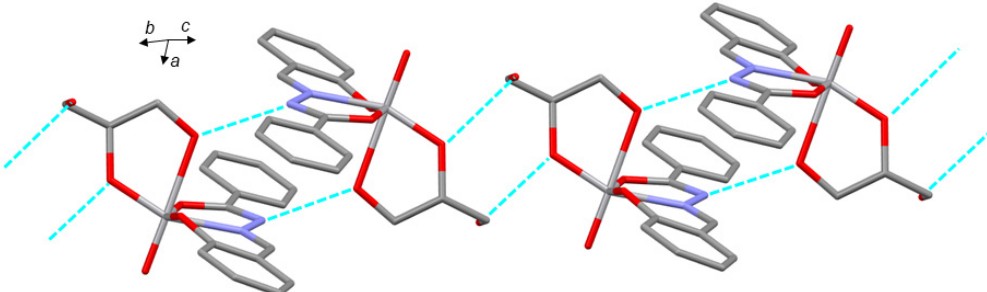

**Figure 5.** Supramolecular arrangement of **8** via intermolecular interactions (MERCURY view, adapted from [22]). Hydrogen atoms are omitted for clarity (dark grey—vanadium, blue—nitrogen, red—oxygen, grey—carbon, white—hydrogen). Hydrogen bonds are shown by light blue dotted lines.

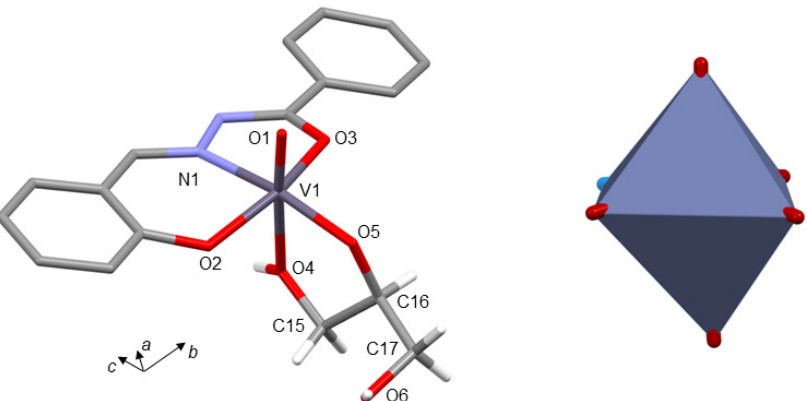

**Figure 6.** Molecular structure (**left**) and geometry atom (**right**) of **8** (MERCURY representation, adapted from [22]). Hydrogen atoms are omitted for clarity, except for those of the glycerolato ligand (colour code: dark grey—vanadium, blue—nitrogen, red—oxygen, grey—carbon, white—hydrogen).

During the same period and using a similar approach, the same group isolated another specimen of vanadium(V) glycerolate in the form of single crystals, characterised as [VO(L)(H$_2$gly)] (**9**) (L = salicylaldimine of glycine) by X-ray crystallographic analysis [23]. Complex **9** was synthesised with excellent yield by reacting [VO(L$^2$)(H$_2$O)] with glycerol in methanol. Crystals were grown by slow evaporation of a methanolic solution and crystalised in the same crystal system and space group as **8** (monoclinic, $P2_1/n$). As in the case of **8**, the vanadium atom of **9** is hexacoordinated describing a distorted octahedral VO$_5$N coordination. The ligand L is tridentate, through two oxygen atoms and one nitrogen atom, generating the formation of two five- and six-membered rings, respectively (Figure 7). The glycerolato ligand is bidentately bonded to the vanadium atom in a V(*O,O*) coordination mode, via two oxygen atoms, one alkoxidic, the other alcoholic. However, none of the [H$_2$gly]$^-$ hydrogen atoms could be precisely located during structural analysis, but the lengths of the V−O bonds clearly corroborate their respective character, i.e., one V−O(alkoxidic) bond [V−O(6) = 1.795(4) Å] and one V−O(alcoholic) bond [V−O(5) = 2.314(5) Å]. The lengths are comparable to those observed for the crystals of **8** in which [H$_2$gly]$^-$ is similarly coordinated. (Table 3). In terms of supramolecular assembly, the uncoordinated alcoholic oxygen atom, located on the pendant arm of [H$_2$gly]$^-$, is in intermolecular hydrogen interaction with two adjacent molecules. The distances measure 2.611(1) and 2.657(13) Å, contributing to the propagation of infinite sheets (Figure 8).

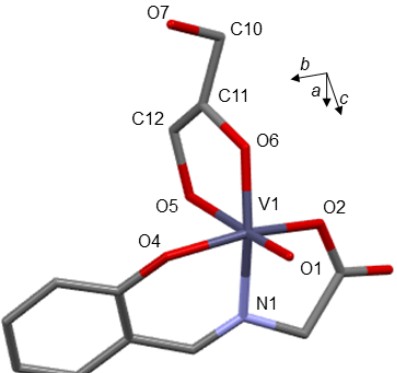

**Figure 7.** Molecular structure of **9** (MERCURY representation, adapted from [23]). Hydrogen atoms are omitted for clarity, (colour code: dark grey—vanadium, blue—nitrogen, red—oxygen, grey—carbon, white—hydrogen).

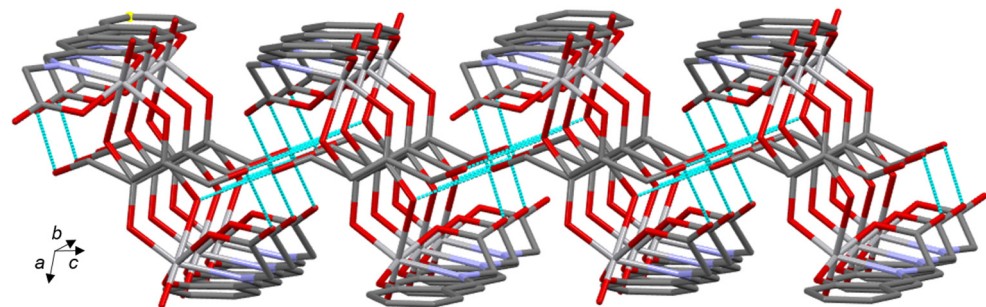

**Figure 8.** MERCURY representation showing the supramolecular arrangement of **9** in the crystal lattice (adapted from [23]). Hydrogen bond interactions are shown by light blue dotted lines (colour code: dark grey—vanadium, blue—nitrogen, red—oxygen, grey—carbon, white—hydrogen).

In 2010, in the frame of studies on the design of suitable single-molecule magnets (SMM [29]), Powel's group reported the synthesis and characterisation of a decanuclear aggregate characterised as $[Mn(II)_2Mn(III)_2Dy(III)_6(\mu_3\text{-}OH)_2(Hgly)_4(H_2gly)_2(PhCO_2)_{16}(H_2O)_2]\cdot 10CH_3CN$ (**10**). Compound **10** was prepared by mixing glycerol, $DyCl_3$, $MnCl_2\cdot 4H_2O$, benzoic acid and $NaN_3$ in acetonitrile, in a molar ratio of 2:1:1:5:6 [30]. Brownish crystals grew after one week at room temperature. The authors described the astonishing structure of 10 as two $\{Mn(II)Mn(III)Dy(III)_2(\mu_3\text{-}OH)(\mu_3\text{-}OR)_3\}$ heterocubane units linked by a central $[Dy(III)_2(PhCO_2)_4]^{2+}$ paddle-wheel dimer (Figure 9, left). It is interesting to note that the skeleton of **10** comprises six glycerolato ligands describing two distinct coordination modes. Two of them are singly deprotonated ($[H_2gly]^-$), while the other four are doubly deprotonated ($[Hgly]^{2-}$). As shown on the right-hand side of Figure 9, each $[H_2gly]^-$ ligand is directly involved in the structure of one of the $Dy_2Mn_2O_4$ heterocubane by providing one of the oxygen atoms [O(19) in the figure], triply-bridging. The two outer hydroxyl groups of $[H_2gly]^-$ are also linked to the cubane by coordinating the two dysprosium atoms. The coordination mode of the $[Hgly]^{2-}$ ligands of 10 is described in Section 3.3.

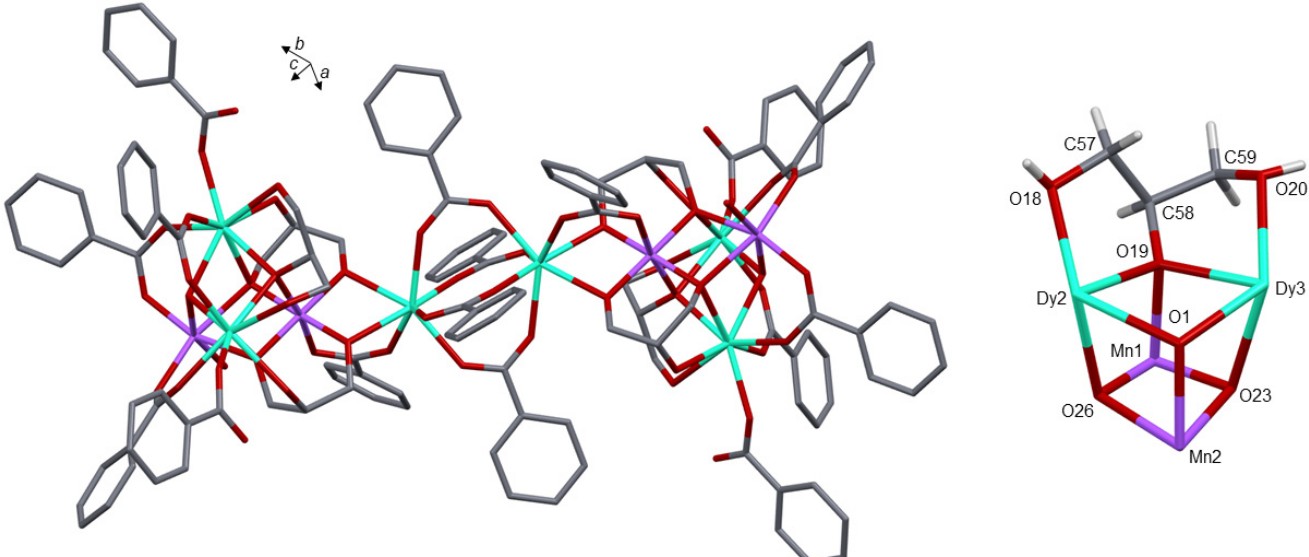

**Figure 9. Left**: global view of the molecular structure of **10** (MERCURY representation, adapted from [30]). Acetonitrile molecules and hydrogen atoms are omitted for clarity. **Right**: focus pointing the coordination of the $[H_2gly]^-$ ligand to the $Dy_2Mn_2O_4$ heterocubane (colour code: green—dysprosium, violet—manganese, red—oxygen, grey—carbon, white—hydrogen).

As part of their work on the design of catalysts for biodiesel production, Schatte's group elucidated in the early 2010s the crystallographic structures of two alkali metal glycerolates, solved as $[Na(C_3H_7O_3)]_n$ (**11**) and $[K(C_3H_7O_3)]_n$ (**12**), and in which glycerol acts as a $[H_2gly]^-$ ligand [31,32]. In both cases, **11** and **12** exhibit polymeric structures,

describing sheet-like organisations that propagate in the directions of the *b*- and *c*-axes, and *a* and *b*, respectively. They crystallise in the monoclinic crystal system, with a $P2_1/c$ space group for **11** and a $C2/m$ for **12**.

The crystals of **11** and **12** were obtained by adding glycerol to hot aqueous solutions of sodium hydroxide and potassium hydroxide, respectively. To be preserved, they must be kept in very basic solutions at room temperature. In $[Na(C_3H_7O_3)]_n$ (**11**), sodium atoms are linked to five oxygen atoms provided by four distinct $[H_2gly]^-$ ligands and display a distorted trigonal bipyramidal geometry. Each $[H_2gly]^-$ ligand is linked to the same sodium atom via an alkoxo group and one hydroxo group (from the secondary carbon atom), leading to the formation of five-membered rings. The second hydroxo group of $[H_2gly]^-$ also interacts with an adjacent sodium atom. Moreover, both OH groups of $[H_2gly]^-$ are also involved in O−H⋯O hydrogen bonds. All these interactions promote the propagation of polymeric sheets (Figure 10).

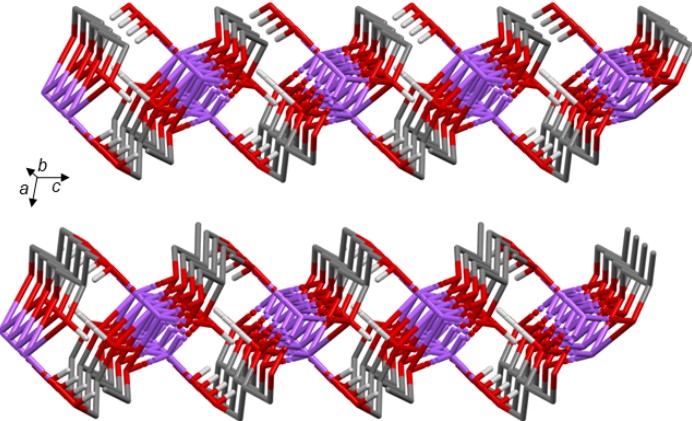

**Figure 10.** Polymeric sheet structure of **11** (MERCURY representation, adapted from [31]). Hydrogen atoms are omitted for clarity (violet—sodium, red—oxygen, grey—carbon, white—hydrogen).

Although it also has glycerolate ligands of the $[H_2gly]^-$ type, $[K(C_3H_7O_3)]_n$ (**12**) exhibits a different mode of coordination compared to **11**. The main difference is that the potassium atom is coordinated by the two hydroxo groups of $[H_2gly]^-$, which leads to a six-membered chelating ring displaying a distorted boat conformation. Each $[H_2gly]^-$ ligand is involved in the coordination with two distinct potassium cations that exhibit a seven-coordination environment resulting from additional K⋯O interactions. A representation of the resulting network is depicted in Figure 11. It consists of parallel polymer sheets propagating along the *a*- and *b*-axes.

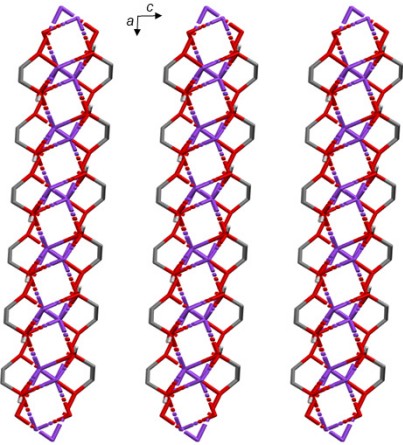

**Figure 11.** Polymeric sheet structure of **12** (MERCURY representation, adapted from [32]). Hydrogen atoms are omitted for clarity (violet—potassium, red—oxygen, grey—carbon, white—hydrogen).

**Table 3.** Comparison of selected structural parameters relevant to the coordination of $[H_2gly]^-$ in crystals **2–12**.

| Crystal | M−O(alcoholic) (Å) | M−O(alkoxide) (Å) | M−O(alcoholic)−C (deg) | M−O(alkoxide)−C (deg) | CSD Entry Deposition Number | Ref. |
|---|---|---|---|---|---|---|
| **2** M = La | 2.592(7) 2.592(8) | 2.383(6) 2.405(7) | 120.0(6) 116.9(7) | 121.2(7) 124.1(7) | VEBYIL 269462 | [21] |
| **3** M = Nd | 2.547(6) 2.526(6) | 2.349(6) 2.330(5) | 116.3(6) 120.8(5) | 124(1) 119(1) | VEBYOR 269463 | [21] |
| **4** M = Gd | 2.472(5) 2.511(5) | 2.300(4) 2.315(5) | 120.6(4) 117.6(4) | 120.5(4) 127.2(4) | VEBYUX 269464 | [21] |
| **5** M = La | 2.598(9) 2.577(8) | 2.410(8) 2.388(9) | 116.1(8) 119.4(7) | 121.0(8) 123.3(8) | VEBZAE 269465 | [21] |
| **6** M = Nd | 2.556(7) 2.531(6) | 2.362(6) 2.341(6) | 116.7(7) 120.5(6) | 117.6(9) 124.5(9) | VEBZEI 269466 | [21] |
| **7** M = Gd | 2.51(1) 2.46(1) | 2.333(9) 2.301(9) | 120.8(9) 117(1) | 126.3(9) 121.1(9) | VEBZIM 269467 | [21] |
| **8** M = V | 2.308(8) | 1.794(6) | 110.5(6) | 123.8(7) | PUSSIF 1239897 | [22] |
| **9** M = V | 2.312(5) | 1.792(4) | 109.7(9) | 121.2(6) | PUGWUJ 1239014 | [23] |
| **10** M = Mn, Dy | 2.394(3) [a] 2.435(4) [a] | 2.383(4) [a] 2.363(5) [a] 2.308(4) [b] | 114.3(4) [c] 114.4(4) [c] | 120.3(3) [d] 118.3(3) [d] 112.5(3) [d] | PUWYIQ 757611 | [30] |
| **11** M = Na | 2.3551(9) 2.4237(9) 2.3462(10) | 2.3163(9) 2.4243(10) | 110.30(6) 119.44(6) 120.20 (7) | 99.17(7) 132.80(7) | VUYFOL 781197 | [31] |
| **12** M = K | 2.7726(16) 2.8160(15) 2.8576(15) | 2.690(2) 3.211(2) | 100.13(11) 113.60(12) 124.89(12) | 106.22(15) 154.98(16) | IJIWOO 811145 | [32] |

[a] Dy−O bond; [b] Mn−O bond; [c] Dy−O−C angle; [d] Mn−O−C angle.

### 3.3. $[Hgly]^{2-}$ Coordination Mode of Glycerolato Ligand to Metal Centres

In 1983, Hambley and Snow reported the X-ray crystallographic analysis of zinc(II) monoglycerolate (**13**) [33]. Crystals of **13** were obtained by heating glycerol and zinc oxide at 220 °C. They are monoclinic with a $P2_1/c$ space group. Zinc atoms are coordinated with five oxygen atoms from three distinct glycerolato ligands. Their geometry can be described as trigonal bipyramidal. Each glycerolate is bonded to three zinc atoms. The remaining non-coordinated hydroxyl group O(3)H has a hydrogen bonding interaction with the O1 atom [O(3)H···O(1) = 2.541(4) Å]. The O(1) and O(2) alkoxides act as bridging ligands between the zinc atoms (Figure 12). From a supramolecular point of view, the resulting organisation can be compared to the stacking of independent layers with strong interactions (Figure 13). Such a structure is easily cleavable, giving zinc glycerol lubricating properties [34]. In terms of applications, zinc glycerolate is also recognised as a green and effective catalyst in the transesterification of soya oil with methanol to produce fatty acid methyl esters [35].

In the early 1970s, Slade, Radoslovich and Raupach solved the X-ray structure of cobalt(II) monoglycerolate, $Co[C_3H_6O_3]$ (**14**), which is iso-structural to the zinc derivative **13** [24]. Crystals of **14**, magenta in colour, grew from a mixture of glycerol and cobalt acetate that was heated for two days at 140 °C. They crystallise in the monoclinic $P2_1/c$ space group. The chemical structure is similar to that described above for the zinc(II) monoglycerate **13**. The values for interatomic angles and distances are also comparable (Table 4). The cobalt atom is located in a trigonal bipyramidal environment coordinated to five oxygen atoms. Although the hydrogen atoms could not be located at the time due to the accuracy of the X-ray data, the singly deprotonated $[Hgly]^{2-}$ nature of the glycerolate ligand is supported both by an O···O distance of 2.59 Å, typical of a hydrogen interaction, and by characteristic absorption bands recorded by infrared spectroscopy anal-

ysis [$\nu(O-H)$ = 3410 cm$^{-1}$, $\nu(O-H\cdots O)$ = 2510 cm$^{-1}$, $\delta(O-H\cdots O)$ = 1945 cm$^{-1}$]. To our knowledge, the crystal structure of **14** corresponds chronologically to the first structural resolution of a metal glycerolate.

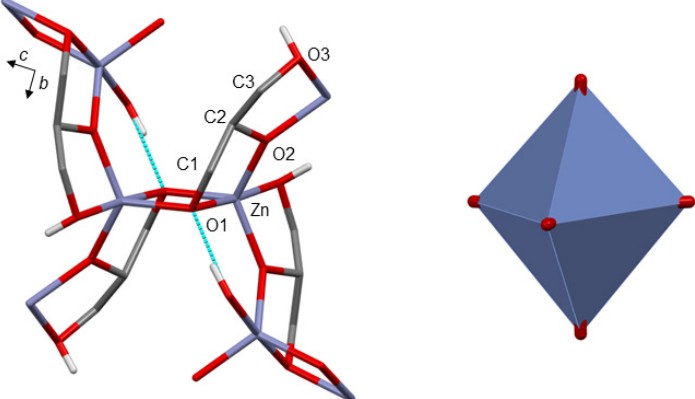

**Figure 12.** Molecular structure (**left**) and metal geometry (**right**) of **13** (MERCURY representation, adapted from [33]) showing dimer formation (hydrogen bonds are shown by blue dotted lines). Hydrogen atoms are omitted for clarity, except OH function of the glycerolato ligand (colour code: dark grey—zinc, red—oxygen, grey—carbon, white—hydrogen).

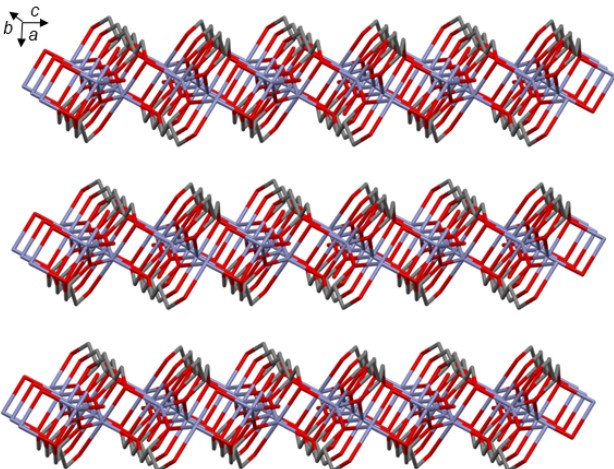

**Figure 13.** Polymeric sheet structure of **13** (MERCURY representation, adapted from [33]). Hydrogen atoms are omitted for clarity (colour code: dark grey—zinc, red—oxygen, grey—carbon).

During the 1980s, the structural resolution of metal glycerolates continued to attract considerable interest. In 1987, Keller and Riebe published the crystal structure of lead(II) monoglycerolate, revealing a polymeric organisation in its solid state, defined as [Pb(C$_3$H$_6$O$_3$)]$_n$ (**15**) [36]. The compound crystallises in the monoclinic space group $P2_1/c$. The authors reported two methods for obtaining single crystals of **15**: either using a sealed ampoule containing a mixture of lead oxide and distilled glycerol and reducing the temperature from 80 °C to room temperature, or from an alkaline solution of plumbate containing glycerol. When this solution is exposed to an atmosphere of acetic acid, colourless needle-shaped crystals are deposited after a few days. Structurally, the lead atom occupies the top of a tetragonal pyramid whose base is occupied by four oxygen atoms from three glycerolate ligands with Pb–O distances in the range of 2.24 to 2.60 Å. Two hydroxyl groups from two [Hgly]$^{2-}$ ligands also interact with the lead atom via two O(H)··Pb bonds [3.03(3) and 3.08 Å], leading to a coordination described by the authors as [4+2]. The presence of the bridging oxygen atoms O(2) and O(3), linked to separate lead atoms, promotes the propagation of a zigzag polymer chain along the *a*-axis (Figure 14). In the crystal

lattice, the chains are linked together through intermolecular hydrogen bonds involving the OH groups of [Hgly]$^{2-}$ ligands of one chain with the lead atoms of neighbouring chains (Figure 15). The thermal stability of **15** was also determined, revealing a decomposition temperature of 236 °C with the formation of Pb–O.

Very recently, as part of investigations into metal alkoxides, Ruck's group published a much faster synthesis route, which produces compound **15** with a 94% yield [37]. The method described involves mixing Pb(OAc)$_2$·3H$_2$O with glycerol, and then adding an aqueous solution of NaOH. The mixture was then heated for five minutes at 140 °C under reflux conditions. The authors obtained a powder pattern comparable to the pattern calculated from the crystal structure published in 1987 by Keller and Riebe [36], as well as a degradation temperature for **14** of around 230 °C. In addition, they confirmed by infrared spectroscopy the presence of broad stretching bands resulting from the presence of the OH groups of the glycerolato ligands.

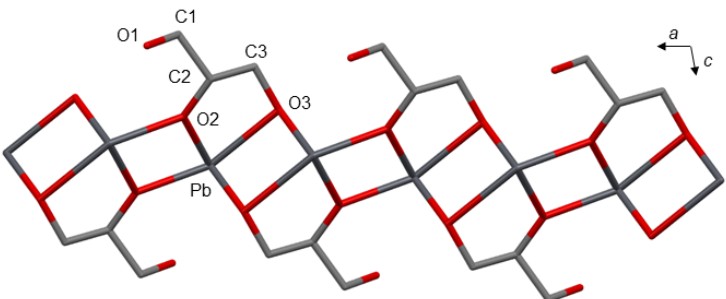

**Figure 14.** Polymeric structure of **15** (MERCURY representation, adapted from [36]). Hydrogen atoms are omitted for clarity (colour code: dark grey—lead, red—oxygen, grey—carbon).

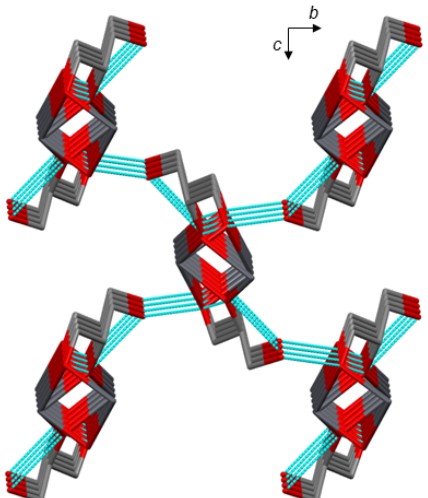

**Figure 15.** MERCURY representation showing the network formed by hydrogen bonds (light blue dotted lines) between the chains of **15** (adapted from [36]).

In 1988, Wild and coworkers reported the crystal structure of (1,2-C$_6$H$_4$(PPhMe)$_2$)Pt(1,2-glycerolate)·2MeOH (**16**) resulting from treatment at 20 °C and in a mixture of benzene–methanol of [Pt(OMe)$_2$\{1,2-C$_6$H$_4$(PMePh)$_2$\}] with one equivalent of glycerol [38]. In the solid state, the platinum atom, which occupies a classical square planar geometry, is doubly chelated by one (1,2-C$_6$H$_4$(PPhMe)$_2$) ligand and one doubly deprotonated glycerolato ligand. The result is the formation of two five-membered metallacycles. In addition, the remaining hydroxyl group of [Hgly]$^{2-}$ is in intermolecular hydrogen interaction with an alkoxo oxygen of a neighbouring complex molecule [O(3)H···O(1) = 2.666(8) Å] forming a centrosymmetric dimer aggregate (Figure 16).

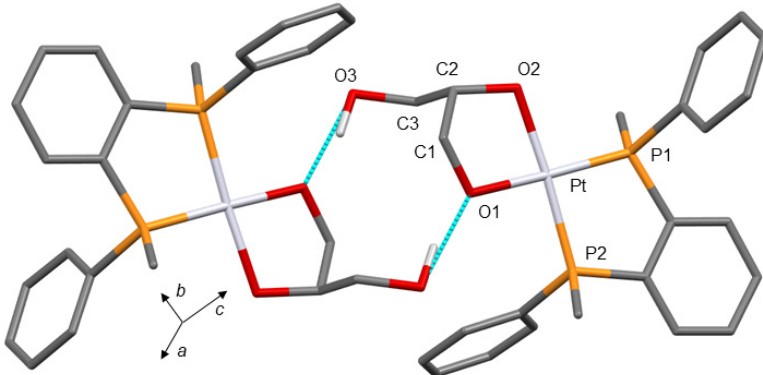

**Figure 16.** Molecular structure of **16** (MERCURY representation, adapted from [38]) showing dimer formation (hydrogen bonds are shown by blue dotted lines). Methanol solvate molecules and hydrogen atoms are omitted for clarity, except for those of the glycerolato ligand (colour code: dark white—platinum, yellow—phosphorus, red—oxygen, grey—carbon, white—hydrogen).

In 1997, as part of a study devoted to polyol metal complexes [39], Klüfers and coworkers reported on the characterisation of multinuclear cuprates(II) with deprotonated glycerol as a ligand. Among the complexes described in the study, one prepared from glycerol, copper(II) hydroxide and barium hydroxide, in solution in water, was characterised by single-crystal X-ray diffraction as $Ba_2(ox)[Cu_2(\mu\text{-}OH)_2(Hgly)_2]\cdot 10H_2O$ (**17**) (ox = oxalate), isolated as blue triclinic crystals. Complex **17** contains two glycerolato $[Hgly]^{2-}$ ligands, each chelating a copper atom and a barium atom (Figure 17). However, significantly different distances were observed for Cu−O and Ba−O bonds involving $[Hgly]^{2-}$ (Table 4). The hydroxyl group O(3)H of $[Hgly]^{2-}$ interacts with the barium atom through a long Ba−O distance (2.847(3)Å) and also with a surrounding molecule by hydrogen bonding. The O(2) atom bridges both a copper atom and a barium atom. Interestingly, the authors explain the presence of the two oxalate ligands, which chelate the two barium atoms, by the oxidation of glycerol in the presence of an aqueous alkaline copper solution. They refer to work dating back to 1936 by Traube and Kuhbier, who had already reported on the oxidation of polyols under such reaction conditions [40].

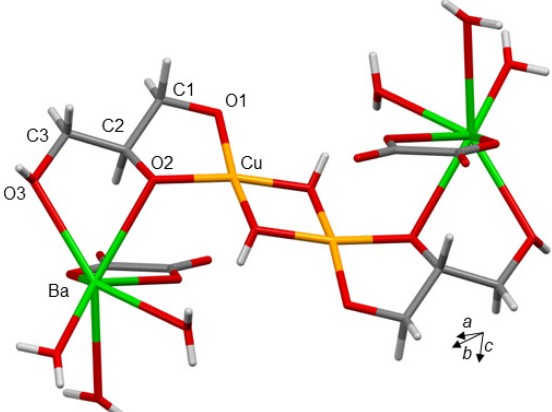

**Figure 17.** Molecular structure of **17** (MERCURY representation, adapted from [39]) showing dimer formation (hydrogen bonds are shown by blue dotted lines). Water solvate molecules are omitted for clarity, (colour code: green—barium, orange—copper, red—oxygen, grey—carbon, white—hydrogen).

In 1999, Klooster and Voss reported the single-crystal X-ray structure determination of three platinum(II) mononuclear complexes being galactitolate, glycerolate and erythritolate ligands [41]. These compounds were synthesised in glovebox conditions, in $CH_2Cl_2$, by reacting (dppp)Pt(CO₃) (dppp = 1,3-bis(diphenylphosphino)propane) with galactitol, glycerol and erythritol, respectively [42]. Suitable single crystals of sugar alcoholate complexes were obtained by slow vapour diffusion of $CH_2Cl_2$ solutions of complexes layered by ether.

With regard to the structure of the glycerol derivative, (dppp)Pt(II)(1,2-glycerolate) (**18**), the platinum atom describes a square planar geometry, bis-chelated, on one side by a dppp ligand and on the other, by [Hgly]$^{2-}$, leading to the formation of two rings with six and five members, respectively (Figure 18). The ring of the dddp chelate displays a flattened boat conformation, while the ring resulting from the coordination of the glycerolate dianion exhibits a twist conformation. Although the hydrogen atom could not be precisely located, the authors support the presence of a free hydroxyl group on the glycerolato ligand [O(3)], which is corroborated by the O(2)···O(3) distance [2.65(3) Å], indicating the presence of an intramolecular hydrogen bond. The structural parameters determined for **18** are comparable to those for complex **15**, although a slight difference is pointed out by the authors for the value of the angle P−Pt−P [87.28(8)° for **15**, 92.0(2)° for **18**].

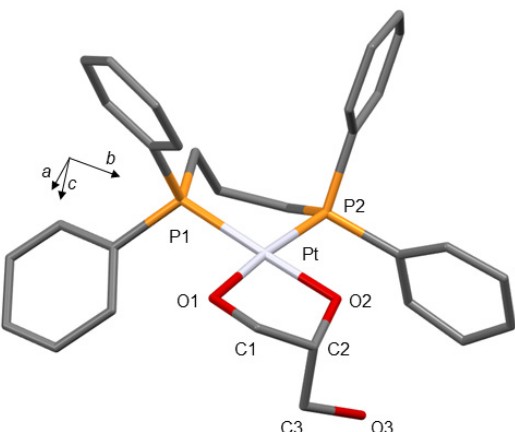

**Figure 18.** Molecular structure of **18** (MERCURY representation, adapted from [41]) showing the presence of two metallacycles. Hydrogen atoms are omitted for clarity (colour code: dark white—platinum, Yellow—phosphorus, red—oxygen, grey—carbon).

As part of their research into synthesising high spin Mn(II)/Mn(III) clusters with magnetic behaviour (for single molecule magnetism application), in 2008, Powell and coworkers reported the synthesis and isolation of a Mn$^{(III)}_{12}$Mn$^{(II)}_9$ aggregate characterised as [{Mn(III)$_{12}$Mn(II)$_9$($\mu_4$-O)$_8$-(Hgly)$_{12}$($\mu$-1,1-N$_3$)$_6$(OH$_2$)$_6$(N$_3$)$_{1.5}${Mn(II)($\mu$-1,3-N$_3$)$_{4.5}$(OH$_2$)$_{1.5}$]Cl$_4$·7.5H$_2$O (**19**) [43]. From a synthetic point of view, compound **19** was prepared by adding MnCl$_2$·4H$_2$O to a solution of glycerol in methanol. Slow evaporation at room temperature led, after three weeks, to the formation of black prism-shaped crystals. Compound **19** crystallises in the cubic space group $Pa\bar{3}$ and the authors compared its structure to the interlocking of concentric Archimedan polyhedra. The centre of the oxocluster is occupied by a cation Mn(II). Twelve doubly deprotonated glycerolate ligands with the same coordination mode are present in the skeleton of **18**. The deprotonated oxygen atoms of [Hgly]$^{2-}$, in particular O(9) and O(10) as shown in Figure 19, are bridging atoms. They each interact with two manganese atoms. The remaining hydroxyl group of [Hgly]$^{2-}$, O(8)H, is, however, simply connected and is linked to a single Mn atom.

The decanuclear aggregate [Mn(II)$_2$Mn(III)$_2$Dy(III)$_6$($\mu_3$-OH)$_2$(Hgly)$_4$(H$_2$gly)$_2$ (PhCO$_2$)$_{16}$(H$_2$O)$_2$]·10CH$_3$CN (**10**), already described in the previous Section 3.2 [30], also contains four [Hgly]$^{2-}$ ligands in addition to the two [H$_2$gly]$^-$ ligands. The doubly deprotonated ligands participate in pairs in the skeleton of the two Dy$_2$Mn$_2$O$_4$ heterocubanes forming part of the structure of **10**. Their central oxygen atoms, O(23) and O(26), are triply bridging ($\mu_3$), occupying two vertices of the cubane. Each is linked to two manganese atoms, Mn(2) and Mn(1), and to one dysprosium atoms, Dy(3) and Dy(2), respectively. The second alkoxo groups, O(21) and O(25), act as $\mu_2$ ligands, bridging one manganese atom of cubanes and one dysprosium atom of the central dimer, Mn(1) and Dy(1). The hydroxo groups of [Hgly]$^{2-}$, O(22) et O(24), are linked to two dysprosium atoms of cubanes, Dy(2) and Dy(3) (Figure 20).

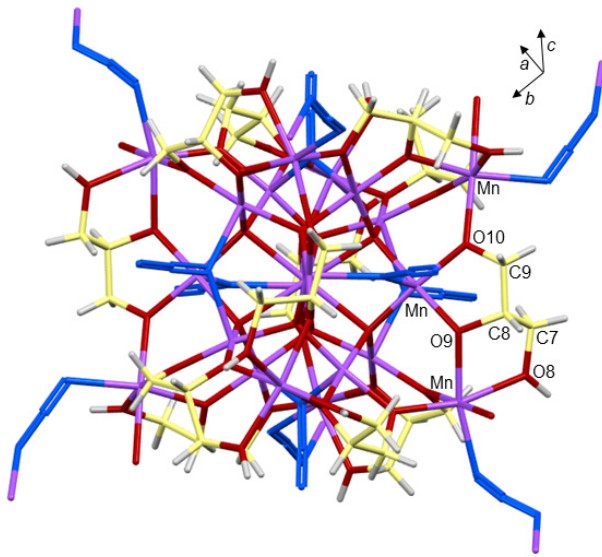

**Figure 19.** Molecular structure of **19** (MERCURY representation, adapted from [43]). The carbon atoms of the twelve [Hgly]$^{2-}$ ligands are shown in pale yellow to highlight their location within the Mn$_{21}$ oxocluster (colour code: violet—manganese, red—oxygen, blue—nitrogen, yellow—carbon, white—hydrogen).

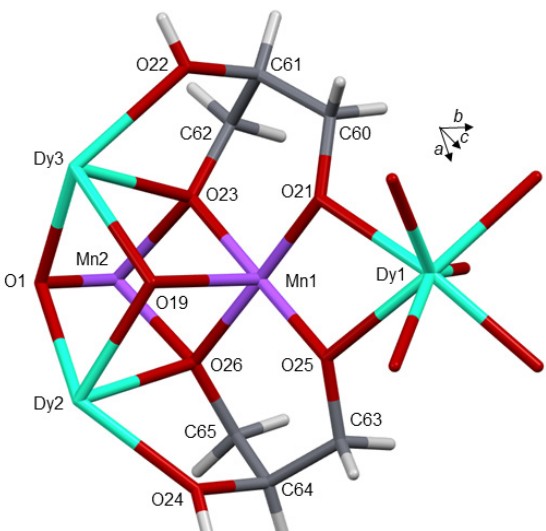

**Figure 20.** Focus showing the coordination of the [Hgly]$^{2-}$ ligands in heterocubane **10** (MERCURY representation, adapted from [30]); colour code: green—dysprosium, violet—manganese, red—oxygen, grey—carbon, white—hydrogen.

With the increase in glycerol production, mainly from the biodiesel industry, glycerol derivatives are also booming and attracting a lot of interest [44]. This is particularly the case of glycerol carbonate (4-hydroxymethyl-1,3-dioxolan-2-one), which has a cyclic carbonate function and a pendant hydroxyl arm. This bifunctional molecule, recognised as harmless and environmentally friendly, is appropriate for a wide range of applications (protic solvent, substitute of ethylene and propylene carbonate, electrolytes for lithium batteries and cosmetic ingredients. . .) [45]. Synthetically, glycerol carbonate is readily available via transesterification reactions by reacting glycerol with linear dialky carbonates, alkylene carbonate and urea [46]. In the late 2000s and based on previous investigations on the direct carbonation of alcohols [47, 48], the research groups of Ballivet-Tkatchenko and Behr investigated the synthesis of glycerol carbonate from CO$_2$ and glycerol using diorganotin(IV) complexes as precatalysts—organotin derivatives are known to be highly reactive towards carbon dioxide, and structural data

reflecting this behaviour had recently been reviewed [49,50]. As part of this work, they demonstrated the possible coordination of glycerol to tin centres by isolating the di-*tert*-Bu₂Sn(1,2-glycerolate) complex (**20**) [25]. Compound **20** was synthesised by heating in toluene, under reflux conditions in a Dean–Stark apparatus, an equimolar mixture of di-*tert*-butyltin oxide and glycerol. Suitable colourless single crystals were obtained by cooling down a hot solution of **20** in either toluene or CHCl₃. The X-ray structure of **20** can be described as an inorganic dimeric skeleton based on a centrosymtric Sn₂O₂ four-membered ring. Each tin atom adopts a distorted trigonal bipyramidal geometry and is chelated by a bidentate 1,2-glycerolate ligand forming a five-membered ring. The two *tertio*-butyl groups of each tin atom are located in the equatorial plane. One of the oxygen atoms of 1,2-glycerolate, O(1), links to Sn, providing the dimeric structure of **20**. The remaining hydroxyl group of [Hgly]²⁻, linked to C(3), is in hydrogen interaction with the oxygen atom O(2) of a neighbouring dimeric unit. The result is the formation of a polymeric chain that propagates along the *c*-axis (Figure 21). In the infrared spectrum (ATR mode), the presence of the hydroxyl group is highlighted by a broad absorption band at 3208 cm⁻¹. However, in terms of reactivity, the authors reported that **20** does not react with CO₂ at ambient conditions, probably resulting in its oligomeric character. The authors also investigated the reactivity of *n*-butyl tin derivatives with glycerol, using *n*-Bu₂Sn(OCH₃)₂ and *n*-Bu₃SnOCH₃ as precursors. Infrared spectroscopy measurements and elemental analyses confirmed the formation of di-*n*-butyltin glycerolate derivatives, but to date, the structures have not yet been confirmed by single-crystal X-ray diffraction analysis.

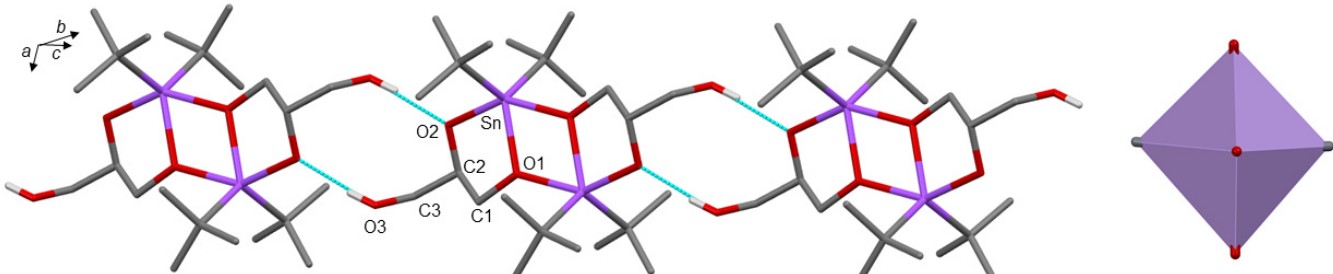

**Figure 21. Left**: molecular structure of **20** (Mercury representation, adapted from [25]) showing dimer formation and propagation of a one-dimensional chain along the *c*-axes, through intermolecular interactions (hydrogen bonds are shown by blue dotted lines). Hydrogen atoms are omitted for clarity, with the exception of the hydroxyl groups of glycerolate ligands (colour code: violet—tin, red—oxygen, grey—carbon, white—hydrogen). **Right**: metal geometry.

The synthesis and characterisation of calcium glycerolates have been the subject of much interest in the last century [51–53]. Today, they are still widely studied as catalysts for transesterification and polymerisation reactions [54,55]. In 2013, Cabeza, Granados and coworkers solved the X-ray structure of Ca(C₃H₇O₃)₂ (**21**) [26]. Synthetically, compound **21** was prepared from fresh CaO obtained by calcining CaCO₃, then heated to 50 °C in a methanol–glycerol mixture in a hermetically sealed flask and under an inert atmosphere. The crystal structure of **21** reveals the presence of isolated tetramers of the formula Ca₄(C₃H₇O₃)₈ whose inorganic core consists of a Ca₄O₄ cubane (Figure 22). Two types of glycerolato ligands decorate the central cube. Four [Hgly]²⁻ ligands chelate the four calcium atoms via two of their oxygen atoms, leading to the formation of five-membered rings. The remaining -CH₂OH group of each [Hgly]²⁻ is oriented towards neighbouring tetramers via hydrogen bond interactions. However, the hydrogen atoms were not located when the structure was solved, but infrared spectroscopy data (DR mode) also confirmed this interpretation (Table 1). In addition, the structure of **21** also includes four [gly]³⁻ ligands whose coordination will be described in Section 3.4. The four calcium atoms of **21** are thus coordinated with seven oxygen atoms, describing a pentagonal bipyramidal geometry.

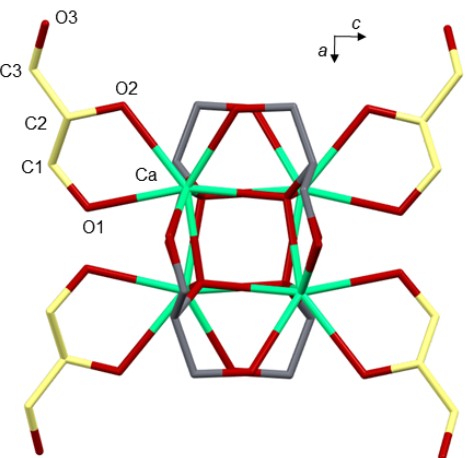

**Figure 22.** Molecular structure of **21** (MERCURY representation, adapted from [26]) highlighting the four [Hgly]$^{2-}$ ligands whose carbon atoms are shown in pale yellow (colour code: green—calcium, red—oxygen, yellow—carbon ([Hgly]$^{2-}$), grey—carbon ([gly]$^{3-}$)).

In 2016, Wei and coworkers published the preparation of unprecedented diol functionalised Anderson-type polyoxometallates (POMs) [27]. In particular, as part of this work, they reported the single-crystal X-ray diffraction of [TBA]$_3$[[CHOH(CH$_2$O)$_2$]CrMo$_6$O$_{18}$(OH)$_4$] (**22**) (TBA = tetrabutylammonium). Compound **22** was obtainded by mixing an aqueous solution of [NH$_4$]$_3$[CrMo$_6$O$_{18}$(OH)$_6$] to a solution of glycerol dissolve in HCl (1 M). After heating at 100 °C for three hours under reflux conditions and the addition of [TBA]Br, a pink crystalline product corresponding to **22** was isolated with a 66% yield. Compound **22** crystallises in the monoclinic *P*2$_1$ space group. The molecular structure reveals the presence of a doubly deprotonated glycerolate ligand, [Hgly]$^{2-}$, attached to the heteropoly [CrMo$_6$O$_{18}$(OH)$_4$]$^-$ anion via four Mo–O bonds and two Cr–O bonds (Figure 23). The oxygen atoms of [Hgly]$^{2-}$ are thus triply bridging ($-\mu_3$), and the resulting interatomic distances are significantly different (Table 4). The secondary alcohol function of glycerol is maintained intact but does not interact supramolecularly. Compound **22** was also characterised by infrared spectroscopy (transmission mode) and mass spectrometry (ESI), in acetonitrile, revealing a mass cluster at *m/z* = 1559.66 assigned to the {(TBA)$_2$[[CHOH(CH$_2$O)$_2$]CrMo$_6$O$_{18}$(OH)$_4$]} moiety. Two other specimen of alkoxo-derivatised Anderson POM clusters were obtained using (HOCH$_2$)$_3$CNH$_2$ and (HOCH$_2$)$_2$(C$_2$H$_5$)CNH$_2$ instead of glycerol.

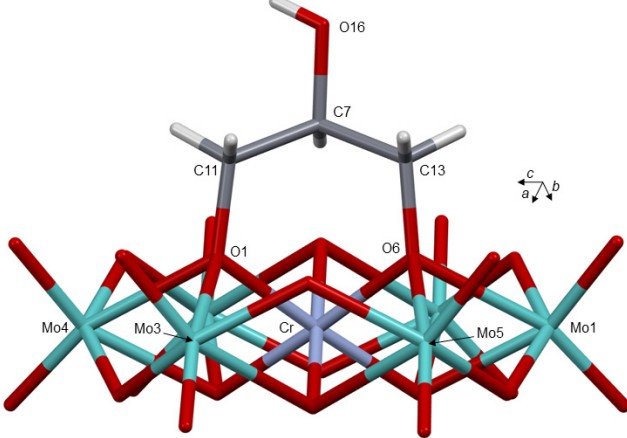

**Figure 23.** Molecular structure of **22** (MERCURY representation, adapted from [27]). TBA cations, water molecules and hydrogen atoms, with the exception of those of ligand [Hgly]$^{2-}$, are omitted for clarity (colour code: violet—chromium, blue green—molybdenum, red—oxygen, grey—carbon, white—hydrogen).

Recently, in 2019, Dybstev and coworkers reported the synthesis and the X-ray crystallographic characterisation of a series of five new zinc(II)-thiophene-2,5-dicarboxylate MOFs referred to as the NIIC-10 series [56]. These buildings are based on 3D porous structures consisting of dodecanuclear zinc(II) carboxylate wheels whose inner walls are decorated with deprotonated polyatomic alcohols (coming from ethylene glycol, 1,2-propanediol, 1,2-butanol, 1,2-pentanediol, and glycerol). Their structures are similar. Among the compounds described in the study, one, characterised as $[Zn(tdc)_6(Hgly)_6(dabco)_3]$ (**23**) (tdc = thiophene-2–5-dicarboxylate, dabco = 1,4-diazobicyclo [2,2,2]octane), was prepared using glycerol. Colourless hexagonal prismatic crystals were obtained by adding glycerol to a mixture of $Zn(NO_3)_2 \cdot 6H_2O$, $H_2tdc$, dabco and DMF (heated in a close vial at 130 °C for 2 days). Compound **23** crystallises in the trigonal $R\bar{3}m$ space group. Glycerol molecules are localised within the MOF channels and are present in the form of deprontonated glycerolate ligands, $[Hgly]^{2-}$, chelating zinc atoms to form five-membered metallacycles. Zinc atoms have two distinct types of coordination geometry, alternately tetrahedral and square-bipyramidal. Interestingly, all $[Hgly]^{2-}$ ligands conserve a pendant -$CH_2OH$ arm, non-coordinated (Figure 24). This particular arrangement, compared by the authors to that of a cyclodextrin, results in great $CO_2/N_2$ and $CO_2/CH_4$ adsorption selectivities, as well as alkali metal cation adsorption properties.

The following year, in 2020, the same research group designed a new family of mesoporous MOFs, referred to as the NIIC-20 series and based on dodecanuclear wheel-shaped carboxylate building blocks $[Zn_{12}(iph)_6(glycol)_6(dabco)_3]$ [57]. Among these new buildings, one, characterised as $[Zn(iph)_6(Hgly)_6(dabco)_3]$ (**24**) was also synthesised from glycerol added to a mixture of $Zn(NO_3)_2 \cdot 6H_2O$, isophtalic acid and dabco in DMF (heated at 130 °C for 48 h). From a structural point of view, the framework structure of **24** is comparable to **23**, describing also a nanocage, and the $[Hgly]^{2-}$ ligands are similarly coordinated to zinc atoms. The discrepancy lies mainly in the carboxylate linkers, tdc for **23** and iph for **24**, which generate geometrical differences. In terms of properties, the NIIC-20 MOFs are considered promising materials for the purification of ethylene from ethane.

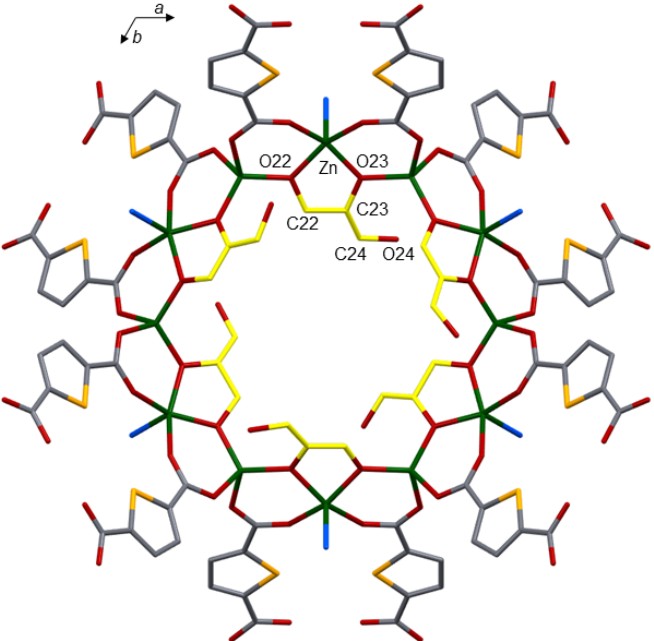

**Figure 24.** Molecular structure of **23** (MERCURY representation, adapted from [56]). The carbon atoms of the six $[Hgly]^{2-}$ ligands are shown in pale yellow to highlight their location inside the $Zn_{12}$ wheel (colour code: green—zinc, red—oxygen, blue—nitrogen, yellow—carbon, white—hydrogen). Hydrogen atom and dabco molecules are omitted for clarity.

**Table 4.** Comparison of selected structural parameters relevant to the coordination of [Hgly]$^{2-}$ in crystals **10** and **13–23**.

| Crystal | M−O(alcoholic) (Å) | M−O(alkoxide) (Å) | M−O(alcoholic)−C (Deg) | M−O(alkoxide)−C (Deg) | CSD Entry Deposition Number | Ref. |
|---|---|---|---|---|---|---|
| **10** M = Mn, Dy | 2.382(6)[a] 2.413(3)[a] | 2.326(4)[a] 2.351(3)[a] 2.515(5)[a] 2.528(4)[a] 1.883(4)[b] 1.895(4)[b] 1.929(4)[b] 1.940(4)[b] 2.284(4)[b] 2.295(4)[b] | 122.3(3)[c] 125.3(4)[c] | 110.5(3)[c] 111.7(3)[c] 119.5(3)[c] 122.9(3)[c] 116.5(3)[d] 117.0(4)[d] 122.2(3)[d] 123.3(4)[d] 126.5(3)[d] 127.2(3)[d] | PUWYIQ 757611 | [30] |
| **13** M = Zn | 2.112(3) | 1.973(3) 1.976(3) 2.009(3) 2.142(3) | 108.5(2) | 97.2(4) 107.1(2) 110.6(2) 115.5(2) 120.0(2) | QQQAZD01 1243918 | [33] |
| **14** M = Co | 1.978 | 1.951 1.971 1.980 2.073 | 106.26 | 109.15 103.67 113.75 124.21 | GLYCCO10 116949 | [24] |
| **15** M = Pb | | 2.24(3) 2.28(3) 2.33(3) 2.60(2) | | 104(2) 117(2) 123(2) 126(2) | FOGXUU 1158502 | [36] |
| **16** M = Pt | | 2.028(5) 2.039(6) | | 107.0(5) 109.8(5) | GETKUL 1166513 | [38] |
| **17** M = Ba, Cu | 2.847(3)[e] | 1.924(2)[f] 1.950(2)[f] 2.821(3)[e] | 114.3(2)[g] | 101.6(2)[g] 108.7(2)[h] 110.2(2)[h] | TIDTOP 127080840 | [39] |
| **18** M = Pt | | 1.985(16) 2.025(15) | | 108(1) 110(2) | HOLGOE 133897 | [41] |
| **19** M = Mn | 2.1744 2.2254 | 1.8751 1.8875 2.1341 | 107.81 113.82 | 111.33 113.72 114.03 | TONBAA 693464 | [43] |
| **20** M = Sn | | 2.0694(17) 2.0860(18) 2.2481(17) | | 111.44(15) 112.14(15) 135.55(15) | HOPKUU 889174 | [25] |
| **21** M = Ca | | 2.506(6) 2.579(6) | | 115.4(4) 118.6(4) | LEYYOF 828033 | [26] |
| **22** M = Cr, Mo | | 1.982(7)[i] 1.988(6)[i] 2.350(6)[j] 2.364(6)[j] 2.383(6)[j] 2.390(6)[j] | | 116.5(5)[k] 118.0(5)[k] 119.2(5)[l] 119.5(5)[l] 120.8(5)[l] 122.7(5)[l] | ZUZVUN 1422707 | [27] |
| **23** M = Zn | | 1.897 1.993(4) 1.993(7) | | 111.4(7) 115.8(8) 119.3 | TOYYEO 1885440 | [56] |
| **24** M = Zn | | 1.89 1.99 | | 110.8 128.0 | WUTHOL 2005258 | [57] |

[a] Dy−O bond; [b] Mn−O bond; [c] Dy−O−C angle; [d] Mn−O−C angle, [e] Ba−O bond, [f] Cu−O, [g] Ba−O−C angle, [h] Cu−O−C angle, [i] Cr−O bond, [j] Mo−O bond, [k] Cr−O−C angle, [l] Mo−O−C angle.

### 3.4. [Hgly]$^{3-}$ Coordination Mode of Glycerolato Ligand to Metal Centres

In 1898, while working on the reactivity of alkali salts with glycerol, Bullnheimer reported the formation of LiCuC$_3$H$_5$O$_3$·6H$_2$O (**25**), characterised as elongated dark blue crystals. They were obtained by mixing glycerol with copper(II) acetate in the presence of

lithium hydroxide, and in a mixture of water and ethanol [58]. Almost 100 years later, in 1993, Klaassen and Klüfers reproduced the reaction and successfully solved the structure of **25** by single-crystal X-ray diffraction [59]. Crystals of **25** crystallise in the trigonal $P\bar{3}c1$ space group. The inorganic framework of **25** is based on a central $Cu_3O_3$ core tricuprate(II) ions exhibiting Cs-symmetry, consisting of a six-membered ring in chair conformation, and alternating oxygen and copper atoms. Three deprotonated glycerolato ligands complete the tricuprate ion structure; each coordinated with two distinct copper atoms. The central alkoxo group of each $[gly]^{3-}$ bridges two copper atoms. The coordination of the three $[gly]^{3-}$ ligands generates six five-membered chelate rings surrounding the central core. Cu(II) ions adopt a square planar geometry (Figure 25).

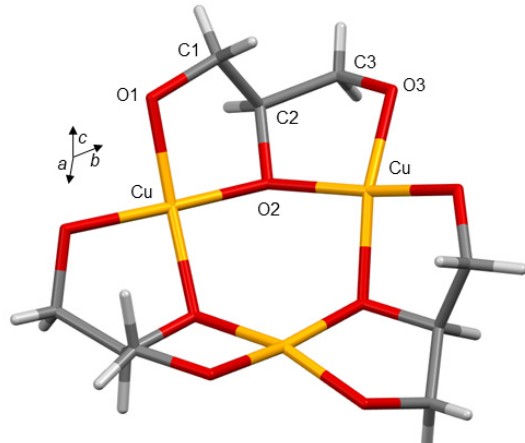

**Figure 25.** Molecular structure of **25** (MERCURY representation, adapted from [59]) (colour code: yellow—copper, red—oxygen, grey—carbon, white—hydrogen). Lithium counter-anions are omitted for clarity.

A few years later, in 1997, Klufers' group isolated two new copper(II) complexes including exclusively deprotonated glycerol as ligands: $Na_3[Cu_3(gl)_3]\cdot7H_2O$ (**26**) and $Na_3[Cu_3(gly)_3]$ $^1/_3NaNO_3$ $10H_2O$ (**27**) [39]. In the same study, the authors also described $Ba_2(ox)[Cu_2(\mu\text{-}OH)_2(Hgly)_2]\cdot10H_2O$ (**17**) which has already been commented on in Section 3.3, being endowed with $[Hgly]^{2-}$ ligands. Compound **26** was prepared by mixing glycerol, copper(II) hydroxide and sodium hydroxide in water giving within two weeks, blue monoclinic crystals. Compound **27** was prepared by replacing copper(II) hydroxide with copper(II) nitrate, according to the method previously described by Bullnheimer [58]. Crystals of **27** were isolated as blue hexagonal plates. The inorganic structures of **26** and **27** are identical and are also based on a central $Cu_3O_3$ tricuprate(II) core comparable to compound **25**. The glycerolate ligands are positioned and coordinated to copper centres in the same way. The main difference is that for **26** and **27**, the $Cu_3O_3$ inorganic ring is capped by a sodium atom which interacts (with distances ranging from 2.3 to 2.9 Å) with oxygen atoms of the central alkoxo groups, O(2), of the $[gly]^{3-}$ ligands (Figure 26).

In 2006, as part of their bioinorganic-oriented work, Klüfers and coworkers demonstrated that the ReI(CO)₃ fragment is a suitable platform for polyol coordination [60]. Among the crystals isolated and described in the study, two included a glycerolato ligand acting as $[gly]^{3-}$: $(DBUH)_2[Re_3(CO)_9(\mu_3\text{-}O)(\mu_3\text{-}gly)]\cdot0.5MeCN$ (**28**) and $(NEt_4)[Re_3(CO)_9(\mu_3\text{-}OMe)(\mu_3\text{-}gly)]$ (**29**). Compound **28** was prepared from an acetonitrile solution containing $(NEt_4)_2[Re(CO)_3Br_3]$ and glycerol, in the presence of 1,4-diazabicyclo[2,2,2]octane (DBU) and a drop of water. The mixture was heated at 85 °C for 6 h. Pale yellow crystals then grew at 4 °C. Concerning **29**, isolated as colourless crystals, the synthesis is similar except that the drop of water is replaced by a drop of methanol. In both compounds, the fully deprotonated glycerolato ligand, $[gly]^{3-}$, is linked to the rhenate(I) trinuclear framework via its three oxygen atoms, each bridging two separate rhenium atoms (Figure 27). In **28**, the Re₃ skeleton is also coordinated by a $\mu_3$-oxo ligand from a water molecule, while in **29** the same position is

occupied by a $\mu_3$-methoxo ligand from a methanol molecule. The $[Re_3(CO)_9(\mu_3\text{-}O)(gly)]^{2-}$ and $[Re_3(CO)_9(\mu_3\text{-}OMe)(gly)]$ moieties were also confirmed by fast atom bombardment mass spectroscopy (FAB-MS), revealing mass clusters at $m/z = 916.8$ and 930.8, respectively. The study also comprises the synthesis and structural characterisation of six other complexes resulting from reactions with three diols ((1*R*,2*R*)-cyclohexane-1,2-diol, anhydroerythritol, (1*S*,2*S*)-cyclopentane-1,2-diol) and three other triols (methyl-*β*-*D*-Ribopyranoside, *L*-threitol, *D*-arabitol).

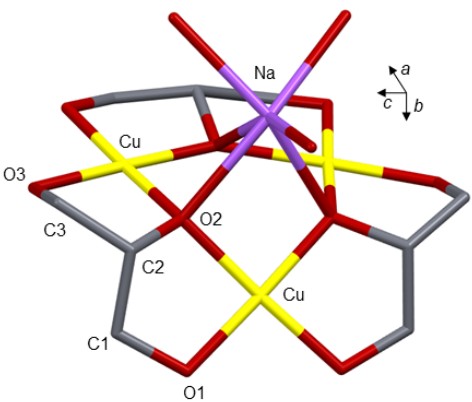

**Figure 26.** Molecular structure of **26** (MERCURY representation, adapted from [39]) (colour code: yellow—copper, violet—sodium, red—oxygen, grey—carbon, white—hydrogen).

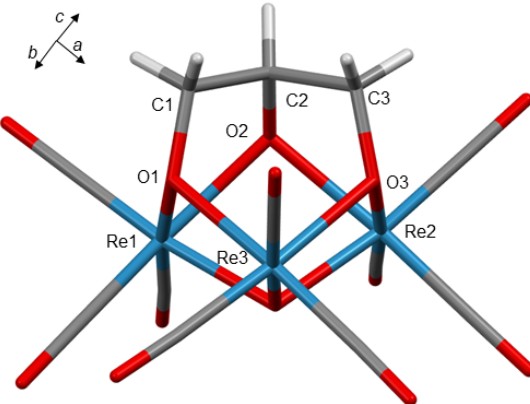

**Figure 27.** Molecular structure of **28** (MERCURY representation, adapted from [60]). (colour code: blue—rhenium, red—oxygen, grey—carbon, white—hydrogen).

In 2012, Raptis and coworkers published the synthesis and characterisation of three new polynuclear Cu(II)-pyrazolato complexes [28]. This type of compound is of interest in terms of its magnetic and electrochemical manipulations [61]. One of the compounds described in this study, which was characterised by single-crystal X-ray diffraction, reveals the presence of two fully deprotonated glycerolato ligands within its structure. $Cu^{II}_8(\mu_3\text{-}\kappa^3,\kappa^2,\kappa^2\text{-}gly)_2(\mu\text{-}3,5\text{-}Me_2\text{-}pz)_8(3,5\text{-}Me_2\text{-}pzH)_2(PhCOO)_2]$ (**30**) was isolated fortuitously by mixing $Cu(OH)_2$, benzoic acid, 3,5-dimethylpyrazole and triethylamine in an acetonitrile solution. The authors explain the presence of glycerolato ligands by the fact that the commercial source of $Cu(OH)_2$ is stabilised by glycerol. Compound **30** grew at room temperature as a blue plate-like crystal crystallising in the $P\bar{1}$ triclinic space group. Compound **30** consists of an octanuclear $Cu_8$ complex whose structure can be viewed as two $Cu_3$ triangles linked by two copper centers. This assembly generates an ellipsoidal ring that hosts two glycerolato ligands ($[gly]^{3-}$) which show comparable coordination modes to copper centres (Figure 28). The oxygen atom of the central alkoxo group, O(1), is triply bridging ($\mu_3$) and linked to three distinct copper atoms, while the other two oxygen atoms, O(2) and O(3) from the two other alkoxo groups, bridge two copper atoms.

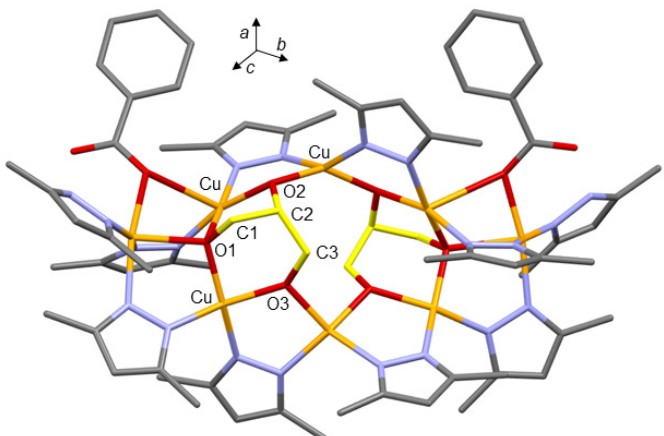

**Figure 28.** Molecular structure of **30** (MERCURY representation, adapted from [28]) highlighting the two glycerolato ligands in yellow. Hydrogen atoms are omitted for clarity (colour code: orange—copper, red—oxygen, blue—nitrogen, grey—carbon, yellow—carbon ([gly]$^{3-}$)).

Recently, in 2019, as part of a study devoted to metal alkoxides, Ruck and coworkers reported the synthesis and characterisation of tin and lead alkoxides of ethylene glycol and glycerol [37]. From glycerol, they isolated, as single crystals, the mixed-valent tin(II,IV) glycerolate, $Sn_5(C_3H_5O_3)_4$ (**31**) and the lead(II) glycerolate, $Pb(C_3H_6O_3)$ (**15**), already described in Section 3.3. Single crystals of **31** were obtained using a PTFE-lined autoclave, heated to 250 °C for 4 h and containing a mixture of tin(II) oxalate and glycerol. Compound **31** crystallises in the tetragonal space group $P4_2/n$ and contains $Sn^{2+}$ et $Sn^{4+}$ ions in an arrangement shown in Figure 29 (left). Glycerol is exclusively present as a triply deprotonated ligand, [gly]$^{3-}$. Each of the three oxygen atoms of [gly]$^{3-}$ is linked to two tin atoms (Figure 29, right). Consequently, the $Sn^{2+}$ atoms are tetra-coordinated and occupy the top of a square-based pyramid, while the $Sn^{4+}$ atoms are octa-coordinated and are located at the centre of triangular dodecahedrons. The authors confirmed the absence of free hydroxyl groups by infrared spectroscopy (ATR mode) and demonstrated the stability of the crystals in air up to 300 °C by DTA-TGA experiments.

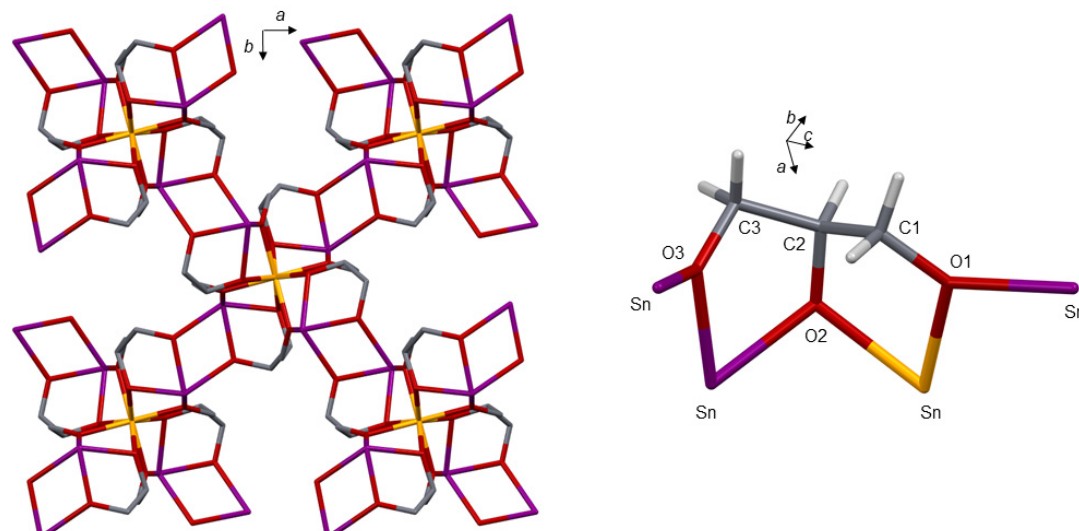

**Figure 29. Left**: Global view of the molecular structure of **31** (MERCURY representation, adapted from [37]). Hydrogen atoms are omitted for clarity (colour code: yellow—tin(IV), violet—tin(II), red—oxygen, grey—carbon). **Right**: Focus pointing out the coordination of the [gly]$^{3-}$ ligand to tin atoms.

In 2022, Kubiak's group reported the synthesis and characterisation of a series of $(OR)_3Sn$-capped trinuclear nickel clusters [62]. The typical structure of this family of

compounds, described as a tin platform, is claimed to favour their reactivity, particularly with small molecules [63]. In this recent study, the authors described the reactivity of [Ni$_3$(dppm)$_3$($\mu_3$-Cl)($\mu_3$-Sn(OEt)$_3$] (dppm = diphenylphosphinemethane) towards ten equivalents of glycerol, leading to the formation of the new cluster [Ni$_3$(dppm)$_3$($\mu_3$-Cl)($\mu_3$-Sn(gly)] (**32**). To our knowledge, this is the most recent example of coordination involving the [gly]$^{3-}$ ligand. Single crystals were obtained using vapour diffusion of diethyl ether into a THF solution of **32** at $-20$ °C. The inorganic framework consists of three nickel atoms supported by three bridging dppm ligands, forming a triangular base capped by a tin atom ($\mu_3$) and a chlorine atom ($\mu_3$). In addition, one molecule of glycerol in its triply deprotonated form, [gly]$^{3-}$, also caps the tin atom (Figure 30). The three alkoxide groups are linked to tin with Sn–O bond lengths of the same order (Table 5). The tin atom is thus hexa-coordinated and its geometry can be described as trigonal prismatic.

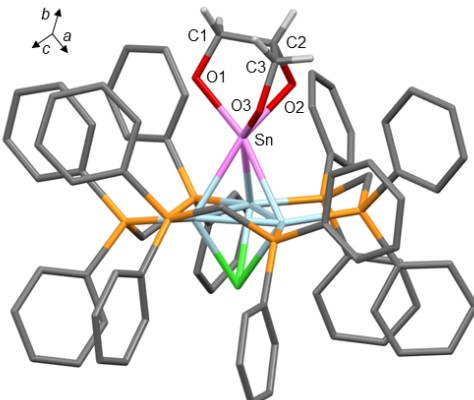

**Figure 30.** Molecular structure of **32** (MERCURY representation, adapted from [62]). Hydrogen atoms are omitted for clarity except for the [gly]$^{3-}$ ligand (colour code: light blue—nickel, pink—tin, green—chloride, orange—phosphorus, red—oxygen, grey—carbon, blue—nitrogen).

Finally, compound **21** was already described in Section 3.3. (relating to [Hgly]$^{2-}$ coordination mode) and exhibits a tetrameric structure defined as Ca$_4$(C$_3$H$_7$O$_3$)$_8$ [26], and also contains four [gly]$^{3-}$ ligands, in addition to the four doubly deprotonated [Hgly]$^{2-}$ ligands. The four [gly]$^{3-}$ ligands are directly involved in the construction of the Ca$_4$O$_4$ cubane, respectively, providing the four oxygen atoms. The central alkoxo groups of [gly]$^{3-}$, O(5), act as $\mu_3$ ligands and are linked to three distinct calcium atoms. The two remaining alkoxo oxygen atoms, O(4) and O(6), are uniquely bonded to two separate calcium atoms, giving rise to two five-membered rings. A total of eight rings are located around the cubane (Figure 31).

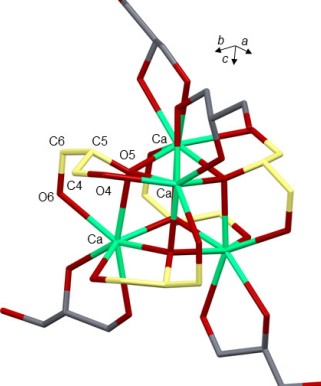

**Figure 31.** Molecular structure of **21** (MERCURY representation, adapted from [26]) highlighting the four [gly]$^{3-}$ ligands whose carbon atoms are shown in pale yellow. The carbon atoms of the four [gly]$^{2-}$ ligands are shown in pale yellow (colour code: green—calcium, red—oxygen, yellow—carbon ([gly]$^{3-}$), grey—carbon ([Hgly]$^{2-}$)).

**Table 5.** Comparison of selected structural parameters relevant to the coordination of $[gly]^{3-}$ in crystals **21** and **25–31**.

| Crystal | M−O(alkoxide) (Å) | M−O(alkoxide)−C (deg) | CSD Entry Deposition Number | Ref. |
|---|---|---|---|---|
| **21** M = Ca | 2.371(6) 2.388(5) 2.446(6) 2.485(5) 2.506(6) 2.510(7) 2.579(6) | 112.3(4) 114.1(5) 115.4(4) 118.0(4) 118.6(4) 120.8(4) 122.1(4) | LEYYOF 828033 | [26] |
| **25** M = Cu | 1.907(7) 1.924(8) 1.950(7) 1.966(6) | 106.8(5) 107.5(7) 108.3(5) 109.5(7) 109.5(6) | JUYKAP 1191635 | [59] |
| **26** M = Cu, Na | 1.891(3) [a] 1.895(4) [a] 1.898(4) [a] 1.909(4) [a] 1.910(4) [a] 1.935(3) [a] 1.936(4) [a] 1.945(3) [a] 1.951(2) [a] 1.955(3) [a] 2.386(3) [b] 2.393(4) [b] 2.405(4) [b] 2.445(4) [b] 2.796(3) [b] 2.895(3) [b] | 105.4(2) [c] 108.0(3) [c] 108.1(3) [c] 108.4(3) [c] 109.3(3) [c] 109.5(3) [c] 109.6(3) [c] 109.7(3) [c] 124.1(3) [d] 132.5(2) [d] 133.1(3) [d] 133.1(3) [d] 136.5(3) [d] 123.7(2) [d] | TIDTIJ 1270839 | [39] |
| **27** M = Cu, Na | 1.895(6) [a] 1.924 [a] 1.933(4) [a] 1.934 [a] 1.940(5) [a] 1.957 [a] 2.351 [b] 2.908 [b] | 106.7 [c] 107.3 [c] 107.8(4) [c] 108.0(4) [c] 108.8 [c] 126.7 [d] 127.9 [d] | TIDTUV 103461 | [39] |
| **28** M = Re | 2.111(5) 2.117(4) 2.127(5) 2.130(5) 2.137(4) 2.168(5) 2.174(5) 2.220(5) | 109.1(5) 120.1(5) 112.8(4) 112.8(4) 106.6(5) 118.2(5) | VEHNUS 292788 | [60] |
| **29** M = Re | 2.131(7) 2.141(6) 2.157(7) | 108.6 110.6(8) 117.1 | VEHPAA 292789 | [60] |
| **30** M = Cu | 1.920(9) 1.923(9) 1.929(6) 1.935(7) 1.940(8) 1.946(8) 1.947(8) 1.952(7) 1.962(8) 1.999(8) 2.020(5) 2.193(8) 2.237(8) | 105.5(7) 107.0(6) 108.4(6) 108.6(6) 111.2(7) 111.6(7) 115.3(6) 118.4(7) 119.3(7) 119.5(7) 120.9(7) 121.2(7) 122.9(7) 126.9(6) | DEGBIC 876485 | [28] |
| **31** M = Sn | 2.111 2.112(3) 2.149(3) 2.228 2.309(3) | 113.7(3) 114.1(2) 115.7 117.0 131.0(3) 132.2(3) | BOLMUN 1939474 | [37] |
| **32** M = Sn | 2.0416(16) 2.0580(15) 2.0645(14) | 94.5(2) 106.7(2) 111.7(2) | NEPGOI 2163770 | [62] |

[a] Cu−O bond; [b] Na−O bond; [c] Cu−O−C angle; [d] Na−O−C angle.

## 4. Conclusions

Glycerol is undoubtedly one of the most versatile bio-based molecules, involved in a wide range of industrial applications and playing a key role in the formulation of many commodity products. The aim of this structural inventory was to compile the single-crystal X-ray structures of metal complexes containing the glycerol molecule and glycerol anions as ligands. It reveals a rich and diversified coordination chemistry. Examples of complexes with glycerol adducts (H$_3$gly) are rather rare. They are exclusively based on the formation of alcoholic M–O bonds with a metal atom. However, once deprotonated, once ([H$_2$gly]$^{2-}$), twice ([Hgly]$^{3-}$) or completely ([gly]$^{3-}$), glycerolato ligands can also initiate alkoxidic M–O bonds with one or more metal atoms, increasing the possibilities for connections. Thus, the binding modes observed between glycerol and metal atoms are eclectic, leading to original species of various nuclearities, ranging from mononuclear complexes to polynuclear clusters and polymeric frameworks. To date, we have identified 32 X-ray structures of metal complexes involving glycerol or glycerolates (summarised in Table 6). In many cases, the study of glycerol coordination was intended as a model study, with the aim of extending it to more challenging polyols, such as carbohydrates. The glycerolate group can be the sole ligand, or it can also be combined with other types of ligands (such as phosphines, carbonyl groups and N- and O-donors...), greatly increasing the design and structural possibilities. The nature of the metallic elements involved in the crystal structures is also very varied, as shown by the red underlined boxes in the figure below (Figure 32), covering all the blocks of the periodic table (alkali, alkaline earth, lanthanide, transition and main group metals). However, it is clear that investigations to uncover new compounds can be pursued, with undoubtedly bright prospects (many elements still lack glycerol-associated structures). This is all the more relevant as, beyond the structural aspect, metal glycerolates are still attracting a great deal of interest in topical areas, such as catalysis, advanced materials and biological activity. Glycerol may be considered an "old-molecule", since it has been known and used for many years, but in our view, it remains a relevant and promising ligand in the field of coordination chemistry.

**Table 6.** Summary of the coordination modes observed in the crystal structures described in this survey.

| Coordination Modes of Glycerol and Glycerolato Ligands | Crystals |
|---|---|
| H$_3$gly | 1, 2, 3, 4, 5, 6, 7 |
| [H$_2$gly]$^-$ | 2, 3, 4, 5, 6, 7, 8, 9, 10, 11, 12 |
| [Hgly]$^{2-}$ | 10, 13, 14, 15, 16, 17, 18, 19, 20, 21, 22, 23, 24 |
| [gly]$^{3-}$ | 21, 25, 26, 27, 28, 29, 30, 31, 32 |

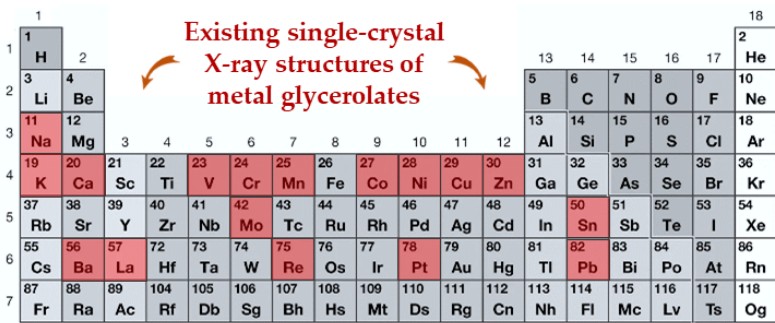

**Figure 32.** Elements (in red) for which X-ray structures of glycerolates were identified and described herein.

**Funding:** L.P. thanks the Centre National de la Recherche Scientifique (C.N.R.S-France) and the University of Bourgogne for financial support.

**Conflicts of Interest:** The author declare no conflicts of interest.

**Abbreviations**

(in order of appearance)
$H_3$gly = glycerol
IUPAC = International Union of Pure and Applied Chemistry
M-glycerolate = metal glycerolates
CSD = Cambridge Structural Database
$[H_2gly]^-$ = monodeprotonated glycerol
$[Hgly]^{2-}$ = doubly deprotonated glycerol
$[gly]^{3-}$ = triply deprotonated glycerol
M = metal
NMR = nuclear magnetic resonance spectroscopy
MOF = metal-organic framework
py = pyridine
FTIR = Fourier transform infrared spectroscopy
ATR = attenuated total reflection
DR = diffuse reflectance
acac = acetylacetonate
$H_2$pd = propane-1,3-diol
ox = oxalate
dppp = 1,3-bis(diphenylphosphino)propane
POMs = polyoxometallates
TBA = tetrabutylammonium
ESI = electrospray ionisation
NIIC = Nikolaev Institute of Inorganic Chemistry
tdc = thiophene-2,5-dicarboxylate
dabco = 1,4-diazobicyclo[2,2,2]octane
DMF = dimethylformamide
iph = isophtalate
DBU = 1,4-diazabicyclo[2,2,2]octane
FAB-MS = fast atom bombardment mass spectrometry
3,5-$Me_2$-pzH = 3,5-dimethylpyrazole
PTFE = polytetrafluoroethylene
DTA-TGA = differential thermal analysis—thermogravimetric analysis
dppm = diphenylphosphinemethane
THF = tetrahydrofuran

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
