# Peer review of "Glycerol as Ligand in Metal Complexes—A Structural Review"

_crystals, doi:10.3390/cryst14030217_

Round 1

Reviewer 1 Report

Comments and Suggestions for Authors

This review paper provides a comprehensive overview and discission of crystal structures of glycerolate-based metal complexes. A scientific significance and practial relevance of the topic are beyond doubt, as also unveiled by the author in the introduction. All sections are written clearly and can provide for anyone a sufficient understanding of glycerol coordination chemistry. The work can be obviously recommended for publication after considering some minor notes: 

1. It would be convenient to cleary state (line 92) the date of final use of CSD Web interface for the preparation of this review. 

2. Line 114: Ni3(1,3,5-benzenetricarboxylate)2(py)6(glycerol)

3. Line 256: a closing bracket should be not in the superscript

4. Line 261: It looks incorrect to call the dimeric block based on six-oxygen-coordinated metal ions as "paddle-wheel". I would recommend to erase this description, or to provide some literature references for similar use of "paddle-wheel" term. 

5. I suggest author to add a small table, summarizing separately the overall counts of the reported glycerolate structures with different coordination modes. Alternatively, Scheme 2 can be appended by such numbers. 

6. It is necessary to review (or maybe just to mention) an apparently missed structure [10.1002/anie.202008132]. 

Author Response

Dear Reviewer,

First, I'd like to thank you for taking the time to review my manuscript. Thank you for your corrections and suggestions!

Please find my answers below (in blue):

  1. It would be convenient to cleary state (line 92) the date of final use of CSD Web interface for the preparation of this review.

The following sentence was added in line 96 page 3:

"The last request to prepare this survey was carried out in December 2023".

  1. Line 114: Ni3(1,3,5-benzenetricarboxylate)2(py)6(glycerol)

The correction has been made in the revised manuscript (page 4, line 119).

  1. Line 256: a closing bracket should be not in the superscript

The correction has been made in the revised manuscript (page 11, line 262).

  1. Line 261: It looks incorrect to call the dimeric block based on six-oxygen-coordinated metal ions as "paddle-wheel". I would recommend to erase this description, or to provide some literature references for similar use of "paddle-wheel" term.

These are the terms employed by Powell and coworkers  describing in the original publication (Dalton Trans. 2010, 39, 4740) the structure of compound 10. For this reason and although it may seem inappropriate, I would like to keep them as they are in the manuscript.

  1. I suggest author to add a small table, summarizing separately the overall counts of the reported glycerolate structures with different coordination modes. Alternatively, Scheme 2 can be appended by such numbers.

As suggested, a summary table has been included in the conclusion (page 37).

Reviewer 2 Report

Comments and Suggestions for Authors

The article is a review of structurally characterized complexes with glycerol. In addition, the authors provide information about the methods of synthesis and the physicochemical properties of the compounds.

The article will be of interest to readers of Crystals. There are a number of comments that can be made:

1)      It is necessary to proofread the text for typos. For example, there is a few mistakes in the sentence “Recently, in 2019, Dybstev and coworkers reported the synthesis and the X-ray crystallographic characterization of a series of five new zinc(II)-thiophene-2,5-dicarboxylate… ”.

2)      “Their structures are isostructural”. – needs rephrasing. Either “Structures are similar” or “Complexes are isostructural”.

3)      During the narrative, it is necessary to indicate how the type of coordination of glycerol changes during the transition from a protonated molecule to each of the deprotonated forms.

4)      Are all structures known with glycerol discussed by the authors? It would be useful to indicate the total number of structures known to date.

5)      In my opinion, more specific comparative information about the change in the function of the glycerol ligand upon its deprotonation should be added to the conclusions.

Author Response

Dear Reviewer,

First, I'd like to thank you for taking the time to review my manuscript. Thank you for your corrections and suggestions!

Please find my answers below (in blue):

1)      It is necessary to proofread the text for typos. For example, there is a few mistakes in the sentence “Recently, in 2019, Dybstev and coworkers reported the synthesis and the X-ray crystallographic characterization of a series of five new zinc(II)-thiophene-2,5-dicarboxylate… ”.

Mistakes in lines 586-588 page 25 have been corrected.

2)      “Their structures are isostructural”. – needs rephrasing. Either “Structures are similar” or “Complexes are isostructural”.

Their structures are isostructural” has been changed by “Structures are similar” in the revised manuscript (page 25, line 591).

3)      During the narrative, it is necessary to indicate how the type of coordination of glycerol changes during the transition from a protonated molecule to each of the deprotonated forms.

I suggest inserting the following sentence page 3, line 83, just before the Scheme 2:

While the glycerol adduct is coordinated to the metal only by one or more alcoholic M–O bonds, deprotonated ligands can also involve alkoxide-type bonds to connect to one or more metal centres, which in many examples favours the formation of polynuclear species.”

4)      Are all structures known with glycerol discussed by the authors? It would be useful to indicate the total number of structures known to date.

I have added the following sentence to the conclusion (page 37, line 776):

To date, we have identified 32 X-ray structures of metal complexes involving glycerol or glycerolates as ligands (summarized in Table 6)”.

5)      In my opinion, more specific comparative information about the change in the function of the glycerol ligand upon its deprotonation should be added to the conclusions.

As suggested, the conclusion has been completed, with the following sentence inserted on page 37, line 769.

“Examples of complexes with glycerol adducts (H3gly) are rather rare. They are exclusively based on the formation of alcoholic M–O bonds with a metal atom. However, once deprotonated, once ([H2gly]2−), twice ([Hgly]3−) or completely ([gly]3−), glycerolato ligands can also initiate alkoxidic M–O bonds with one or more metal atoms, increasing the possibilities for connections.”

Reviewer 3 Report

Comments and Suggestions for Authors

In this manuscript, structural characterization based on crystallographic structures previously published in the literature and added to the Cambridge Structural Database are discussed, along with different modes of coordination of glycerol and glycerolates with metals. The manuscript contains a diverse coordination chemistry. However, I believe that some points should be improved before publication. These points are presented below.

·       I recommend that you insert a general index with all the abbreviations in the text, including the linkers described.

·    Put the name and acronym for the metal-organic frameworks (MOFs) in standard form (page 4, line 12). Next, use these acronyms, MOFs, throughout the text. (Line 114, Page 4). It is recommended to read the entire text.

·     Pag 1, line 16. The glycerol molecule is also a metabolite and is generally abbreviated as GLY in the literature. Therefore, I recommend changing the abbreviations (H3Gl, H2Gl, HGl…) to a more standardized form (H3gly, H2gly, Hgly…).

·    Pag 1, line 20. Is radiocrystallographic structure the official term? I don't know the term. Check to see if it exists or if the authors made a mistake.

·  Pag 3, line 76. In the sentence: …diffraction, revealing numerous coordination modes to metal centres. Describe the nature and designations of the coordination modes in Scheme 2. It will, in my opinion/suggestion, be more didactic.

·     Pag 3, line 79-80. In the sentence: ….To our knowledge, however, there are only a few examples of coordination metal complexes directly exhibiting the H3gl glycerol adduct. Insert a reference for this sentence.

·      Pag 3, line 92. The Cambridge Structural Database (CSD) web interface version (2017) is cited by the authors as the version that they used; however, more recent versions make reference to 2023. Why did the authors decide to utilize a previous version?

·       Pag 6, Figure 3. Improve image resolution.

·      Pag 9, line 200-201. Insert the illustration of metal geometry that is given in the text, as a suggestion. Octahedral, distorted tricapped trigonal prism, trigonal bipyramidal geometry.

·   Pag 16, line 360. I recommend changing the sentence to formal English:….The authors reported two methods to get single-crystals of 15:….As example: …. The authors reported two methods for obtaining single crystals of 15:…

·      Pag. 37, last figure. The authors highlighted the metals that coordinate with the glycerol molecule; however, reference 31—which presents cobalt as a crystalline structure—is presented in the text. I suggest highlighting this metal in the periodic table.

After considerations, I recommend a publication.

Author Response

Dear Reviewer,

First, I'd like to thank you for taking the time to review my manuscript. Thank you for your corrections and suggestions!

Please find my answers below (in blue):

    I recommend that you insert a general index with all the abbreviations in the text, including the linkers described.

As recommended, a general index listing all abbreviations has been inserted on page 39.

Put the name and acronym for the metal-organic frameworks (MOFs) in standard form (page 4, line 12). Next, use these acronyms, MOFs, throughout the text. (Line 114, Page 4). It is recommended to read the entire text.

The correction has been made on page 4 (lines 117 and 119) and also on page 25, line 588.

Pag 1, line 16. The glycerol molecule is also a metabolite and is generally abbreviated as GLY in the literature. Therefore, I recommend changing the abbreviations (H3Gl, H2Gl, HGl…) to a more standardized form (H3gly, H2gly, Hgly…).

Abbreviations have been modified as suggested in the revised manuscript and in Scheme 2.

Pag 1, line 20. Is radiocrystallographic structure the official term? I don't know the term. Check to see if it exists or if the authors made a mistake.

The term of “radiocrystallographic” has been changed by “diffraction” in page 1, line 20.

Pag 3, line 76. In the sentence: …diffraction, revealing numerous coordination modes to metal centres. Describe the nature and designations of the coordination modes in Scheme 2. It will, in my opinion/suggestion, be more didactic.

The following sentence has been inserted in line 76 page 3:

Glycerol and its deprotonated forms act as monodentate, bidentate, tridentate and bridging ligands (Scheme 2)”.

Pag 3, line 79-80. In the sentence: ….To our knowledge, however, there are only a few examples of coordination metal complexes directly exhibiting the H3gl glycerol adduct. Insert a reference for this sentence.

We added “[20-21]” at the end of the sentence, line 116 page 4 of the revised manuscript.

Pag 3, line 92. The Cambridge Structural Database (CSD) web interface version (2017) is cited by the authors as the version that they used; however, more recent versions make reference to 2023. Why did the authors decide to utilize a previous version?

This is an error; the version used (2023) has been updated in the revised manuscript.

Pag 6, Figure 3. Improve image resolution.

A new figure has been inserted in page 6 of the revised manuscript.

Pag 9, line 200-201. Insert the illustration of metal geometry that is given in the text, as a suggestion. Octahedral, distorted tricapped trigonal prism, trigonal bipyramidal geometry.

As suggested, an illustration of the metal geometry of complex 8 has been inserted in Figure 5, page 9. This was also done for figures 12 and 21.

Pag 16, line 360. I recommend changing the sentence to formal English:….The authors reported two methods to get single-crystals of 15:….As example: …. The authors reported two methods for obtaining single crystals of 15:…

The change was made in the revised manuscript (page 16, line 366).